

# Advances in mapping sub-canopy snow depth with unmanned aerial vehicles using structure from motion and lidar techniques

Phillip Harder[1], John W. Pomeroy[1], and Warren D. Helgason[1,2]

[1]Centre for Hydrology, University of Saskatchewan, Saskatoon, Saskatchewan, Canada
[2]Department of Civil, Geological, and Environmental Engineering, University of Saskatchewan, Saskatoon, Saskatchewan, Canada

*Correspondence to*: Phillip Harder (phillip.harder@usask.ca)

**Abstract.** Vegetation has a tremendous influence on snow processes and snowpack dynamics yet remote sensing techniques to resolve the spatial variability of sub-canopy snow depth are lacking. Unmanned Aerial Vehicles (UAV) have had recent widespread application to capture high resolution information on snow processes and are herein applied to the sub-canopy snow depth challenge. Previous demonstrations of snow depth mapping with UAV Structure from Motion (SfM) and airborne-lidar have focussed on non-vegetated surfaces or reported large errors in the presence of vegetation. In contrast, UAV-lidar systems have high-density point clouds, measure returns from a wide range of scan angles, and so have a greater likelihood of successfully sensing the sub-canopy snow depth. The effectiveness of UAV-lidar and UAV-SfM in mapping snow depth in both open and forested terrain was tested in a 2019 field campaign in the Canadian Rockies Hydrological Observatory, Alberta and at Canadian Prairie sites near Saskatoon, Saskatchewan, Canada. Only UAV-lidar could successfully measure the sub-canopy snow surface with reliable sub-canopy point coverage, and consistent error metrics (RMSE <0.17m and bias -0.03m to -0.13m). Relative to UAV-lidar, UAV-SfM did not consistently sense the sub-canopy snow surface, the interpolation needed to account for point cloud gaps introduced interpolation artefacts, and error metrics demonstrate relatively large variability (RMSE <0.33m and bias 0.08 m to -0.14m). With the demonstration of sub-canopy snow depth mapping capabilities a number of early applications are presented to showcase the ability of UAV-lidar to effectively quantify the many multiscale snow processes defining snowpack dynamics in mountain and prairie environments.

## 1 Introduction

Snow accumulation and melt are critical parts of the hydrological cycle in cold regions (King et al., 2008). To understand these processes there needs to be robust and accurate observation methodologies to measure the depth and density of a snowpack, and its change, across all aspects of the landscape. Unfortunately, traditional remote sensing methods struggle to quantify the spatial distribution of snow at a high enough resolution and accuracy to account for the fine scale interactions between snow and vegetation (Nolin, 2010). Remote sensing conceptually promises the capability to gather this type of data at the spatial scales and extents needed, but the main challenge for snow observations across a heterogeneous landscape is that exposed vegetation and forests obscure the underlying snow surface (Bhardwaj et al., 2016; Nolin, 2010; Tinkham et al., 2014). This





paper seeks to illuminate some of the challenges posed to UAV-based remote sensing of snow depth observations and how UAV-based lidar represents a promising opportunity to overcome this limitation at the small catchment scale (<5 km$^2$).

Capturing the spatial distribution of snowpacks and snowcover at a particular instance provides information about the integrated accumulation and ablation processes up to that point in time. Accurate quantification of snow accumulation and

ablation is needed to improve the understanding of snow hydrology, test process understandings, examine spatial scaling of process interactions (Clark et al., 2011; Deems et al., 2006; Trujillo et al., 2007), and to initialise and/or validate model predictions (Painter et al., 2016). Snow depth, the focus of this paper, is not the variable of ultimate interest for hydrology. Rather, snow water equivalent (SWE) is used for snow hydrology applications (Pomeroy and Gray, 1995). Fully cognisant of this, the focus here is on snow depth, as it is well documented that snow depth varies much more than density (Pomeroy and

Gray, 1995; Shook and Gray, 1996; Jonas et al., 2009; López-Moreno et al., 2013); therefore, improving the accuracy of snow depth observations in a drainage basin is critical to improving the estimation of SWE at and within basin scales.

Snow depth and SWE observations are traditionally collected though *in situ* observations (Goodison et al., 1987; Helms et al., 2008; Kinar and Pomeroy, 2015a; Sturm, 2015). *In situ* approaches, such as snow surveying, rely on manual sampling of snow depths and densities to get SWE. When conducted along landscape-stratified transects the lansdcape-scale SWE can be

estimated (Pomeroy and Gray, 1995; Steppuhn and Dyck, 1974). The challenge for snow survey observations is that they are prone to observer bias, are labour intensive and time consuming, and are often unable to sample all aspects of a landscape such as avalanche zones (Kinar and Pomeroy, 2015a). Nonetheless, snow surveying is a proven approach to quantify SWE and has been operationalised across many regions. The practice has historical precedence and has created many long-term records which are a valuable data source (Goodison et al., 1987; Helms et al., 2008). Other point observations, such as snow pillows

(Coles et al., 1985), acoustic sensors (Kinar and Pomeroy, 2009; 2015b), and passive gamma sensors (Smith et al., 2017) are valuable automated data sources, but suffer from location/elevation bias -- as demonstrated by the SNOTEL network in the western United States (Molotch and Bales, 2006). In particular, measurements of snow in forest clearings will provide a much greater snowpack than would be found under the adjacent canopy (Pomeroy and Gray, 1995) and so are not suitable for snow hydrology calculations or model validations even though they are often used for just such purposes. Other techniques need to

be developed to capture the small-scale spatial variability of snow-vegetation interactions to advance our process understandings and validate the next generation of distributed snow models.

Remote sensing approaches have shown promise to evaluate snow depth in open areas. Airborne-lidar and UAV Structure from Motion (UAV-SfM) approaches have been proven to provide snow depth mapping abilities when differencing snow-covered (hereafter snow) and snow-free (hereafter ground) Digital Surface Models (DSM). Lidar, an active sensor, emits a

pulse of light and detection of the reflected pulse results in a point cloud of a scene with a consistent quaility point cloud regardless of flight characteristics, wind conditions, or solar illumination. A clear benefit of lidar is that multiple returns per pulse can be observed with points within the canopy and at the underlying surface. In contrast UAV-SfM uses a passive RGB sensor where data quality is not actively controlled. This results in variable image quality because: inconsistent solar illumination influences image exposure; wind gusts influence platform stability leading to blurry images and inconsistent



overlap; and surface heterogeneity means that some areas of the domain will have more key points--points automatically detected and matched in multiple images (Westoby et al., 2012)--leading to variably in the quality of the SfM solution (Bühler et al., 2016; Harder et al., 2016; Meyer and Skiles, 2019).  So while SfM can provide similar quality error metrics in open areas the quality will vary between flights as conditions change, whereas lidar will be more consistent.  Reported snow depth accuracy in open environments, expressed as root mean square errors (RMSE), varies from 0.08 m to 0.60 m for airborne-lidar

(DeBeer and Pomeroy, 2010; Harpold et al., 2014; Painter et al., 2016; Tinkham et al., 2014), 0.17 to 0.30 m for airborne-SfM (Bühler et al., 2015; Meyer and Skiles, 2019; Nolan et al., 2015), and 0.02 to 0.30 m for UAV-SfM (Harder et al., 2016; Vander Jagt et al., 2015; De Michele et al., 2016). A notable challenge is that the presence of exposed vegetation, especially dense forest, confounds SfM solutions and obscures airborne-lidar bare ground extractions which are needed for fine scale differencing of DSMs to evaluate snow depths or snow depth changes (Bhardwaj et al., 2016; Deems et al., 2013; Harpold et

al., 2014). Terrestrial laser scanning (TLS) is another approach for observing high-resolution snow depth data which has been used to develop an understanding of snow depth distributions and for validating other snow depth observation methods (Currier et al., 2019; Egli et al., 2012; Grünewald et al., 2010; Mott et al., 2011). However, TLS has had limited contributions to furthering understanding of snow processes in forested areas as they are restricted to visible open terrain and forest edges (Currier et al., 2019).

Most applications of remote sensing for observing snow processes have focussed on open environments. However, vegetated portions of those same environments can play a large role in landscape-scale snow hydrology. For example, wetland vegetation accumulates deep snowdrifts and so has an exaggerated influence on snow accumulation processes in prairie environments (Fang and Pomeroy, 2009). Similarly, forests constitute large fractions of the mountain domain (Callaghan et al., 2011; Troendle, 1983) and have very different snow processes than found in open environments (Pomeroy et al., 2002). Snow-

vegetation interactions are complex (Gelfan et al., 2004; Hedstrom and Pomeroy, 1998; Harder et al., 2018; Musselman et al., 2008; Parviainen and Pomeroy, 2000; Pomeroy et al., 2001) and involve both snow interception by the canopy and wind redistribution to forest edges. In dense forests vegetation leads to interception and subsequent sublimation of snow resulting in an overall decrease in accumulation (Hedstrom and Pomeroy, 1998; Parviainen and Pomeroy, 2000; Reba et al., 2012; Swanson et al., 1986). In open environments, such as the prairies, tundra and alpine, wind redistribution of snow leads to a

decrease in snow depth in exposed erodible areas and an increase in snow accumulation in aerodynamically rough surfaces or sheltered areas that act as snow sinks – this includes forest edges (Essery et al., 1999; Fang and Pomeroy, 2009; Liston and Hiemstra, 2011; Pomeroy et al., 1993; Schmidt, 1982). Much of the understanding of snow-vegetation interactions is based on snow surveys, which are limited in scale and extent. Thus approaches to systematically and efficiently quantify these dynamics across a drainage basin accounting for topographic and vegetation heterogeneity are needed to further develop and test our

process understandings.



## 1.1 Research Questions and Objectives

The overall motivation of this work is to understand how snow depth, as well as the processes driving its accumulation and ablation, varies across the complex vegetated landscapes. Better tools are needed to measure snow at scales that resolve snow-vegetation interactions, which can involve individual trees and small forest gaps. So the specific objectives in this manuscript 100 are to: 1) evaluate the ability of UAV-lidar versus UAV-SfM to quantify snow depth in open and vegetated areas, and 2) articulate challenges and opportunities for UAV's to map sub-canopy snow depth.

## 2 Data and Methods

### 2.1 Sites

Several sites from western Canada, which represent a range of surface condition and snow climates, were selected to test the 105 ability of the UAV-lidar and UAV-SfM to sample snow depth in open and vegetated areas.

Fortress Mountain Snow Laboratory (hereafter Fortress), in Kananaskis AB (50.833, -115.220), is a research basin operated by the University of Saskatchewan's Centre for Hydrology in support of mountain hydrology research. The 5 km$^2$ catchment's elevation ranges from 2000 m to 2900 metres above sea level (m.a.s.l.). Field observations for this paper focussed on the Fortress Ridge (Figure 1a) which spans an open alpine environment, a larch treeline zone near 2200 m.a.s.l., and a mixed 110 lodgepole pine and subalpine fir forested slope to the valley bottom at 2000 m.a.s.l. (Schirmer and Pomeroy, 2019). The area was developed as an alpine ski resort in the 1960's, currently a limited-use ski operation without snowmaking, and some open ski runs remain through some of the slopes of interest. Strong winds result in substantial redistribution of snow by blowing snow in this environment (Aksamit and Pomeroy, 2018)

Two study areas in the Canadian Prairies were tested in this study. Both sites provide examples of cropland with hummocky 115 terrain subject to significant blowing snow redistribution (Figure 1bc). Windblown snow from upland areas of short vegetation is often transported to lower elevation wetland depressions where it is effectively trapped by wetland vegetation. One site was located southeast of Saskatoon, SK (51.941 N, -106.379 W), hereafter Clavet, with the other site north of Saskatoon, SK (52.694 N, -106.461 W), hereafter Rosthern. The main difference between prairie sites was that Rosthern received more snowfall and developed a deeper snowpack than Clavet in winter 2019. Where results from both sites are aggregated, they are 120 collectively referred to as Prairie hereafter.



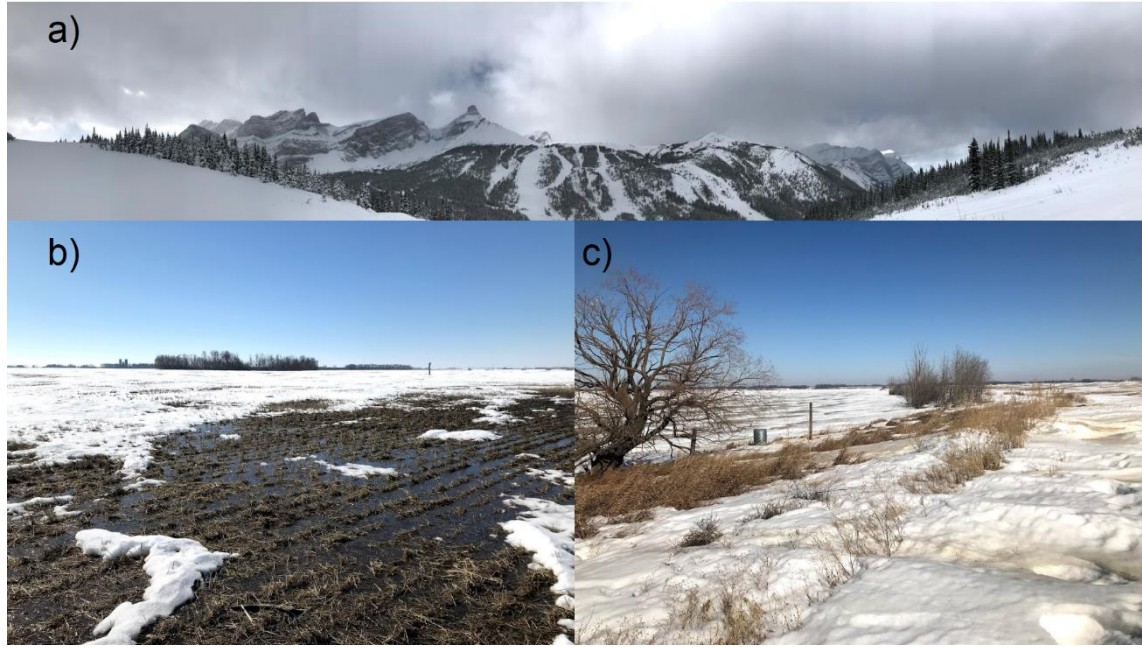

**Figure 1: a) Fortress Mountain Snow Observatory in Kananaskis, Alberta Canada, b) Clavet and c) Rosthern Prairie study locations in Saskatchewan Canada. Data collection was centred on Fortress Ridge (ridgeline in middle of photograph) an area of high topographic variability and variability between dense forests and clearings. The Clavet scene highlights the tall dense grass and wetland vegetation of a wetland and agricultural land transitions. The Rosthern scene highlights the low vertical relief and isolated woodlands amongst cultivated fields.**

## 2.2 Data Collection

### 2.2.1 Lidar System

The UAV-lidar system was comprised of a Riegl miniVUX-1UAV lidar sensor, integrated with an Applanix APX-20 Inertial Measurement Unit (IMU), and mounted on a DJI M600 Pro UAV platform (Figure 2a). The miniVUX1-UAV utilises a rotating mirror to provide a 360-degree line scan with a measurement rate of 100 KHz and up to 5 returns per shot with a 15 mm precision. The APX-20 provides positional accuracy of <0.05m in horizontal and <0.1m in vertical dimensions with a 200Hz sampling rate and 0.015 degree and 0.035 degree accuracy in roll/pitch and heading, respectively. The payload, 5 kg, approaches the maximum capacity of the M600 Pro platform so flight parameters to maximise mapping efficiency were set to 7 m/s ground speed, 100 m flight altitude above the surface, with parallel flight lines 80 m apart. Flight times are conservatively limited to 15 minutes. The generated UAV-lidar point clouds have densities of approximately 75 points per square metre (pt m$^{-2}$).

### 2.2.2 Structure from Motion systems

Coincident surface mapping with SfM used imagery collected by EbeeX or Ebee+ fixed wing UAV platforms with SODA RGB cameras from Sensefly (Figure 2b). The longer flight times, up to 70 minutes, associated with a lightweight payload on



a fixed wing platform allowed for efficient mapping of large areas. Overlap parameters were generally 80% for the longitudinal and 65% in the lateral axes. Flight altitudes of 120 m above the surface provided a ground sample distance of 2.8 cm with the SODA camera, which was used on both EbeeX and Ebee+ platforms. The generated UAV-SfM point clouds have densities of ~ 110 pt m$^{-2}$.

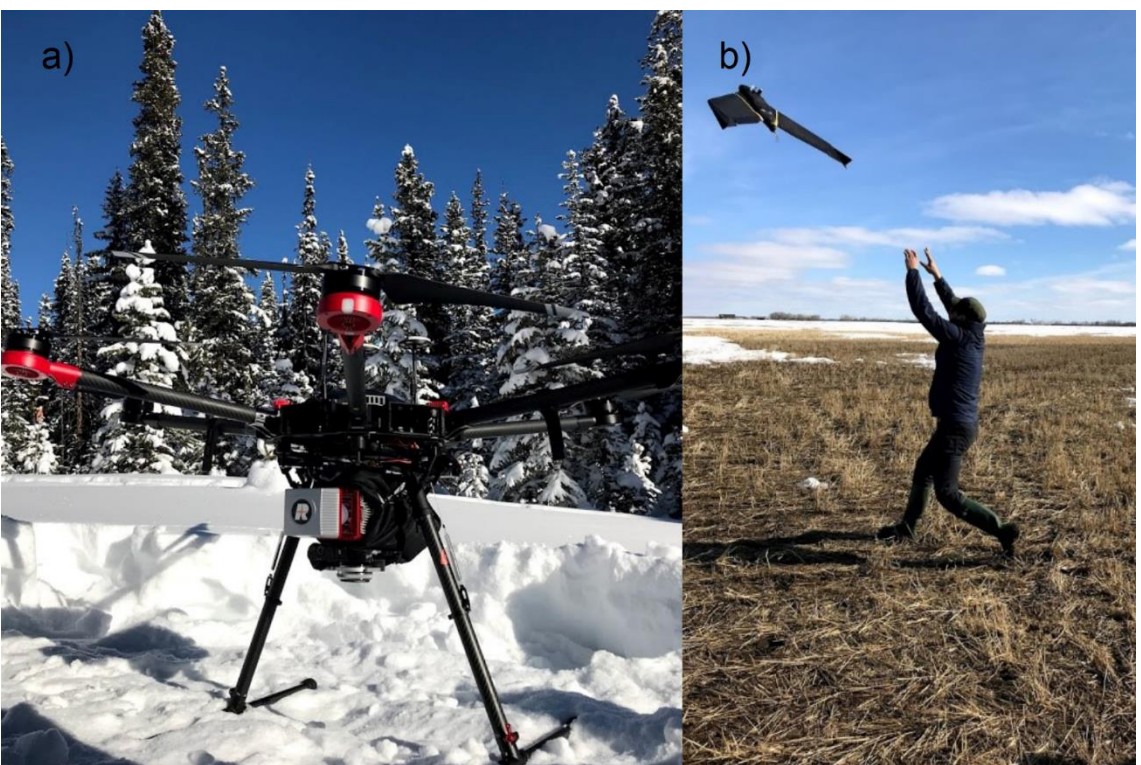


**Figure 2: UAV-lidar platform: Riegl miniVUX1-UAV mounted on DJI M600 Pro (a) and UAV-SfM platform: Sensefly EbeeX (b).**

### 2.2.3 Ground Validation Surveys

The assessment of snow depth accuracy used coincident surveys of surface elevation points with differential Global Navigation Satellite System (GNSS) surveys and manual measurement of snow depths with a ruler. The intention of the surveys was to
validate the spatially distributed snow depth retrievals therefore transects were random within the survey areas and selected in a manner for the surveyor(s) to efficiently sample the greatest variety of vegetation types and gradients. A Leica GS16 base/rover kit provided a real-time-kinematic (RTK) survey solution that allows surveying of points to an accuracy of < ±2.5cm. Post-processing of the GNSS data used the Canadian Geodetic Survey of Natural Resources Canada Precise Point Positioning (PPP) online tool (https://webapp.geod.nrcan.gc.ca/geod/tools-outils/ppp.php) to define an absolute base station
location. Post-processing adjustment of the GS16 rover points to account for the PPP base station location used the Leica Infinity software (version 2.4.1.2955).





### 2.2.4 Campaigns

To assess the accuracy of these methods as well as provide insight into the snow distribution evolution over periods of time 19 surveys were conducted over the course of September 2018 to April 2019. These are summarised by date, surface condition,
data collected, and corresponding number of surface points in Table 1.

**Table 1: Summary of data collection campaign, Sept 2018 to April 2019**

| Date (mm-dd) | Surface | Data Collected | Site | Number of Manual Observations |
|---|---|---|---|---|
| 09-07 | ground | lidar | Rosthern | 0 |
| 09-19 | ground | lidar | Fortress | 0 |
| 10-10 | ground | lidar | Clavet | 0 |
| 12-13 | snow | lidar | Clavet | 0 |
| 01-31 | snow | lidar,SfM | Clavet | 51 |
| 02-13 | snow | lidar,SfM | Fortress | 81 |
| 03-11 | snow | lidar | Clavet | 30 |
| 03-13 | snow | lidar,SfM | Rosthern | 111 |
| 03-15 | snow | lidar | Clavet | 35 |
| 03-18 | snow | lidar,SfM | Rosthern | 81 |
| 03-20 | snow | lidar,SfM | Clavet | 69 |
| 03-22 | snow | lidar,SfM | Rosthern | 72 |
| 03-24 | snow | SfM | Rosthern | 0 |
| 03-26 | snow | lidar,SfM | Rosthern | 73 |
| 03-29 | snow | lidar | Rosthern | 77 |
| 04-03 | snow | lidar | Clavet | 0 |
| 04-04 | snow | lidar | Rosthern | 0 |
| 04-09 | snow | lidar | Rosthern | 0 |
| 04-25 | snow | lidar | Fortress | 39 |

### 2.3 Data Processing

Snow depth was quantified as the vertical difference between a bare ground DSM and a bare snow DSM. This approach was taken regardless of whether point clouds or surface models come from lidar scanning or SfM processing. The workflows implemented to produce point clouds and DSMs vary between lidar and SfM approaches (Figure 3).





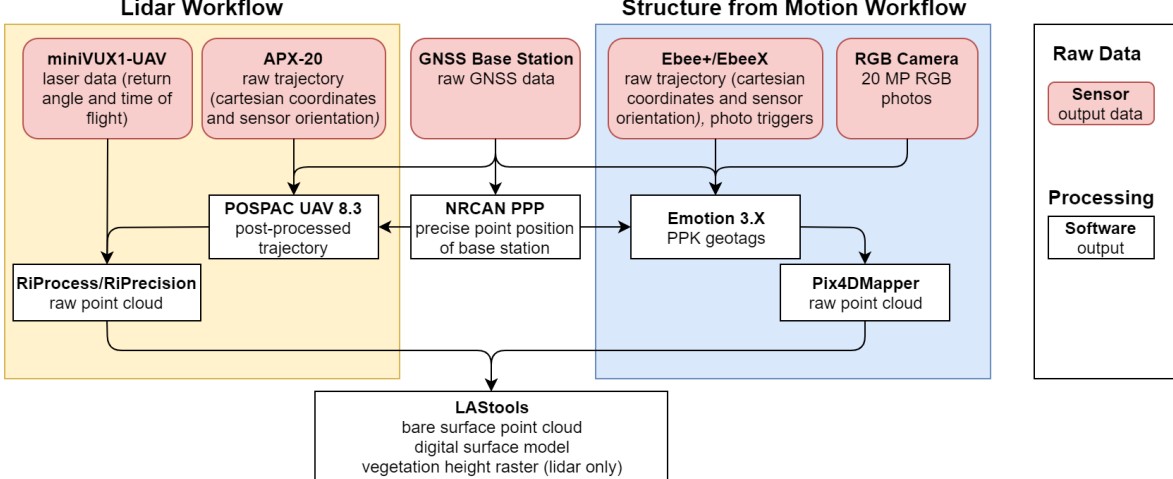

**Figure 3: Data processing workflows for lidar and SfM point cloud generation.**

### 2.3.1 Lidar processing workflow

To generate a georeferenced lidar point cloud several data streams need to be integrated in post processing. The raw high frequency trajectory (x, y, z, pitch, roll, and yaw) information from the APX-20 IMU was post processed with POSPAC UAV software, which includes a post processing kinematic (PPK) correction by integrating base GNSS data from a known point < 2 km from flight area, to provide an absolute sensor position accuracy of <2.5 cm. The post-processed IMU data is merged with the scanner data within the proprietary RiProcess software package to translate the time of flight laser returns to an x, y, and z point. Finally, overlapping scan data is used to optimise the IMU trajectory, laser data accuracy is greater than the post processed IMU trajectory data, to align the scan lines and reduce the noise of the final point cloud within the RiPrecision tool.

### 2.3.2 SfM processing workflow

The UAV-SfM processing workflow begins with associating a high accuracy x, y, and z positon to the images taken. Within the Emotion 3.X software from SenseFly a PPK correction, with raw GNSS data collected at the known point base station, is applied to the photo locations to give geotag accuracies of < ±2.5 cm. The Pix4D Mapper (v 4.3.33) SfM software, with the "3D Maps" default options template, processed the collected imagery and post processed geotags to produce a densified point cloud. Within the study sites a minimum of 5 ground control points (GCP), blue 2 m x 2 m tarps with a white cross, were surveyed with the Leica GS16 rover and integrated into the Pix4D SfM workflow. For further details on how Pix4D implements SfM techniques and more generally the approach to use SfM to map snow depth refer to Harder et al. (2016) and Meyer and Skiles (2019).





### 2.3.3 Point Cloud Processing

The points representing the 'bare' surface, whether that is the snow or ground surface, are of interest for snow mapping. Lidar point clouds comprise of returns from vegetation *and* the snow/ground surface, while UAV-SfM point clouds comprise returns

from vegetation *or* the snow/ground surface and exhibit substantial noise around snow patch edges (Harder et al., 2016). To remove noise and vegetation points a noise removal and bare surface point classification was applied to the point clouds with the LAStools software (Isenburg, 2019). The lidar workflow performed a noise removal followed by a bare surface point classification. For ground surface lidar scans, the height of vegetation (non-ground) points was also calculated. For the UAV-SfM point clouds, the noise removal and bare surface classification follows the workflow of Isenburg (2018).

### 2.3.4 Surface interpolation

A DSM was generated in order to reduce the overall volume of data and to allow for simple surface differencing. The 'blast2dem' tool within the LAStools package generates a seamless triangulated irregular network (TIN) that conforms to the point cloud which is then resampled to a raster (Isenburg, 2019). A spatial resolution of 0.1 m was applied to all DSMs generated.

### 2.3.5 Error Assessment

To assess the accuracy of UAV-lidar and UAV-SfM with respect to observations, a surface based comparison was undertaken. Snow and ground surface values were extracted from the DSM raster cells for locations where a point was manually surveyed and snow depth measured. The snow depth was calculated from the vertical difference between the DSM values for the snow and ground DSM's. The influence of vegetation height on snow depth errors was also considered by segmenting the error

metrics with respect to vegetation height (open <0.1 m, shrub <0.5 m, and trees >0.5 m) derived from the snow-free (ground) UAV-lidar scan. The classified vegetation maps and location of all survey points are visualised in Figure 4. The error metrics employed to assess the differences between observations and estimates included the root mean square error (RMSE), and the mean bias (mb).





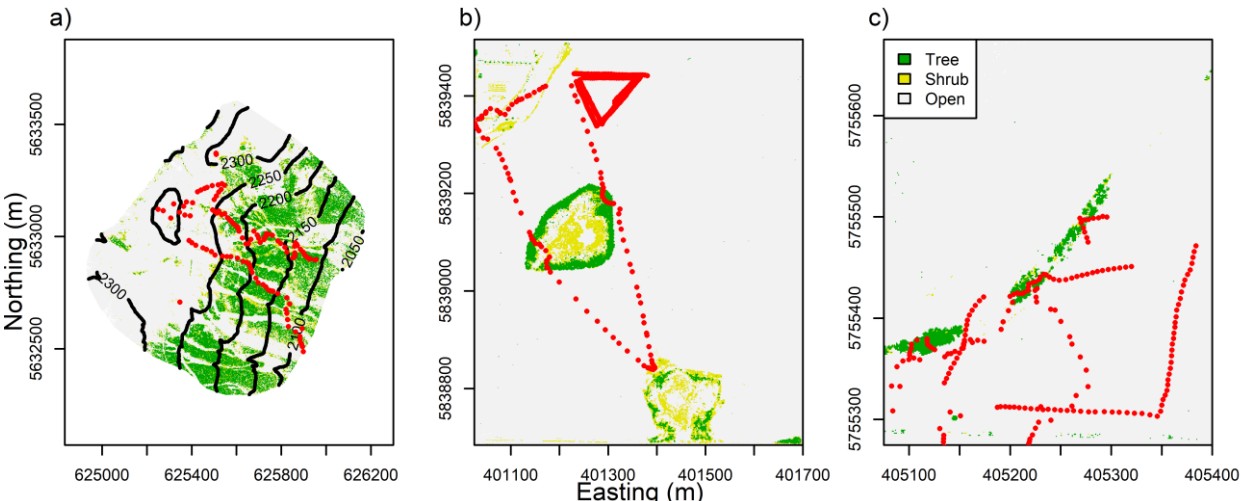

**Figure 4: Fortress a), Rosthern b), and Clavet c) study sites classified by vegetation height derived from snow-free (ground) UAV-lidar into open (<0.5m), shrub (>0.5m and <2m) and tree (>2m) domains. Red points identify locations of manual snow depth survey observations sampled over the course of the data collection campaign. Black lines in Fortress map are 50 m elevation contours.**

### 2.3.6 Point Coverage

The continuity of bare surface point density between UAV-lidar and UAV-SfM methods was quantified in order to interpret how well the respective tools can sense sub-canopy surfaces. All surveys with coincident UAV-lidar and UAV-SfM flights were assessed with the LAStools (Isenburg, 2019) grid_metrics function to classify area with > 1 pt 0.25 $m^{-2}$ and thereafter were summarised as percentage areas of each study site with >1 pt 0.25 $m^{-2}$ with respect to technique. This is a rough metric of DSM quality as it quantifies the relative amount of interpolation needed to translate a point cloud to a continuous surface.

## 3 Results

### 3.1 Accuracy of UAV-lidar versus UAV-SfM

An accuracy assessment comparing the snow depth from UAV-lidar and UAV-SfM techniques to that manually sampled through the RTK ground surveys is shown in Figure 5. UAV-lidar has consistently lower error than UAV-SfM in open environments and mountain vegetation. The exception is prairie shrub vegetation where the UAV-lidar RMSE is slightly larger than UAV-SfM RMSE. The significance of the different relative RMSE values for Prairie shrub vegetation is negligible relative to the much larger differences noted in the other domains. UAV-lidar bias is consistently negative (-0.03 m to -0.13 m), while the UAV-SfM bias is more variable and both positive and negative (0.08 m to -0.14 m).





**Figure 5: Comparison of snow depth observations and UAV-based snow depth estimates. Plots are segmented for points extracted from the point clouds or interpolated surfaces (rows), sites (columns) and observation method (colours).**

The influence of vegetation on snow depth measurement is directly assessed by considering the errors associated with different vegetation classes (Figure 5). When considering UAV-lidar, the errors are worse in the presence of vegetation. Open Prairie and Fortress samples are similar (0.09 m and 0.1 m RMSE respectively), whilst vegetated sites have larger error (0.13 m to 0.17 m RMSE) with no observed dependency upon vegetation class or type. The UAV-lidar is equally successful penetrating the open leaf-off deciduous tree canopy at the prairie sites as the closed needleleaf canopy at the Fortress site. The UAV-lidar RMSE for Shrub and Tree vegetation classes at Fortress and Prairie sites are within 0.04 m. For UAV-SfM the errors differ widely for various vegetation covers. The Open vegetation has a large RMSE range (0.1 m in Prairie and 0.3 m in Fortress





respectively) while vegetation RMSEs range from 0.13 m to 0.33 m. While UAV-SfM reports slightly better metrics than

UAV-lidar in the prairie Shrub case it is within the observational error of RTK survey equipment and reasons will be examined in the discussion. The influence of vegetation type is apparent in the UAV-SFM Tree class errors where the dense needleleaf forest at Fortress has a higher RMSE (0.33 m) than the leaf-off deciduous trees in the prairies (0.2 m). Overall UAV-lidar tends to consistently have lower RMSE's than UAV-SfM which provides confidence in this technique for mapping snow depth sub-canopy.

Snow depth is estimated from differencing the snow and ground DSM. Therefore, the uncertainty of the snow depth is a propagation of the error of both the snow and ground DSMs. To distinguish which DSM may contribute more to the snow depth error, the remotely sensed surface elevations were compared to the surface elevations from the RTK surveys (Figure 6). The UAV-SfM snow surface elevations have errors consistently greater than the corresponding UAV-lidar surfaces at Fortress. In the Prairie snow-surface case, the median RMSE is consistently lower for UAV-SfM than UAV-lidar, but the UAV-SfM

does have more variability in its errors. The ground surface was only available from UAV-lidar for this study so no corresponding UAV-SfM ground surface analysis is available. The snow-free UAV-lidar survey has a consistently higher or more variable RMSE than the snow surfaces (with the exception of the Open Prairie and Open and Tree Fortress UAV-SfM).






**Figure 6: Boxplots of RMSE of UAV estimated and RTK survey surface elevations segmented by surface condition, technique, site, and vegetation classification. The error metrics approach the uncertainty of the RTK survey data +/- 2.5 cm data.**

### 3.2 Point cloud coverage

The quality of a remotely sensed snow depth estimate is directly tied to how much interpolation is required to fill gaps in a

point cloud. The point clouds were classified into areas where >1 pt 0.25 m$^{-2}$ existed for each technique. Examples of this

approach are visualized for the Fortress, Rosthern and Clavet sites on Feb 14, March 18 and March 20, 2019 survey dates in

Figures 7-9 respectively. At the Fortress site (Figure 7b) the large areas of lidar only points (orange) correspond to areas of

forest cover as the UAV-SfM technique could not reliably return surface points whilst the UAV-lidar could. At Fortress UAV-



lidar had > 1 pt 0.25 m$^{-2}$ for 93% of the area of interest versus 54% for UAV-SfM. Considering the Figure 7a transect, the
lack of UAV-SfM points near trees means that the interpolated snow surface does not capture the tree wells, which are sharp
decreases in snow depth around the base of trees, and evident from the UAV-lidar data. The noisy UAV-SFM points in the
middle of the slope challenge the snow surface extraction even without the presence of vegetation leading to an underestimation
of the snow surface. Large areas without UAV-SfM point coverage occurred northwest of the ridge in open areas due to low
surface contrast and surface homogeneity.



**Figure 7: Fortress Ridge (February 14, 2019) study site with an example a) cross section with all points and interpolated vegetation-free surface (lines) for SfM-snow (red), lidar-snow (green) and lidar-ground (blue) surveys. The study area is classified by areas with greater than 1 pt per 0.25 m⁻² in b) with respect to point clouds obtained from UAV-lidar and UAV-SfM techniques. The red inset polygon in b) identifies the area of the orthomosaics displayed in c) with an overlain transparent point type classification. Red line in c) corresponds to the cross section plotted in a).**





The predominantly open nature of the Prairie sites demonstrates a minimal difference in coverage between techniques. The average of 5 coincident flights at Prairie sites gave UAV-lidar a mean coverage of 92% versus 83% for UAV-SfM. As seen in Figure 8 at the Rosthern site, the areas without UAV-lidar points include some wetland shrubs (green areas in Figure 8 b and c), but predominantly are randomly distributed points. In contrast, UAV-SfM is missing points from areas where the snow

surface is very uniform, in vegetated rings around wetlands, and in areas of dense vegetation (orange areas in Figure 8 b and c). These gaps in the point clouds are interpolated during DSM interpolation and therefore will represent areas of greater uncertainty. There was ponded meltwater on the surface of the frozen ground and frozen wetland water surface at the Clavet Site on March 20, 2019, which is responsible for the many areas missing lidar points in Figure 9b. Water is a specular reflector therefore unless the lidar has a nadir perspective water areas will appear as a gap in the point cloud. Fortunately, since water

surfaces are flat, minimal interpolation artefacts remain when generating DSMs from the point clouds if the pond edges are captured. The challenge in the prairies, as seen in Figure 8a and 9a, is in areas of thick but short vegetation (shrub class) where lidar pulses and SfM solutions interpret the vegetation surface as the ground surface and therefore the remotely sensed ground surface, and UAV-SfM and UAV-lidar snow surface are very similar. An additional challenge of UAV-SfM due to challenges in vegetation removal in bare surface generation is that large gaps appear beneath the tall wetland edge vegetation leading to

points, as visualized by transects in Figure 8 and 9, where the estimated UAV-SfM snow surface is below the UAV-lidar ground surface.





Figure 8: Rosthern (March 18, 2019) study site with an example a) cross section with all points and interpolated vegetation-free surface (lines) for SfM-snow (red), lidar-snow (green) and lidar-ground (blue) surveys. The study area is classified by areas with greater than 1 pt per 0.25 m$^{-2}$ in b) with respect to point clouds obtained from UAV-lidar and UAV-SfM techniques. The red inset polygon in b) identifies the area of the orthomosaics displayed in c) with an overlain transparent point type classification. Red line in c) corresponds to the cross section plotted in a).



**Figure 9: Clavet (March 20, 2019) study site with an example a) cross section with all points and interpolated vegetation-free surface (lines) for SfM-snow (red), lidar-snow (green) and lidar-ground (blue) surveys. The study area is classified by areas with greater than 1 pt per 0.25 m⁻² in b) with respect to point clouds obtained from UAV-lidar and UAV-SfM techniques. The red inset polygon in b) identifies the area of the orthomosaics displayed in c) with an overlain transparent point type classification. Red line in c) corresponds to the cross section plotted in a).**



## 4 Discussion

### 4.1 UAV-lidar is more accurate and consistent than UAV-SfM

Snow depth mapping with UAVs has had widespread application in recent years (Bühler et al., 2016; Harder et al., 2016;
Vander Jagt et al., 2015; De Michele et al., 2016). The emphasis has been on using SfM techniques to difference DSMs. One
of the objectives of this work was to consider the snow depth accuracies possible with the current state of the art of UAV-SfM
versus UAV-lidar platforms. What has been demonstrated here is that while there are still errors in UAV-lidar (as with any
measurement), they are smaller and more consistent relative to UAV-SfM. An unavoidable problem for all SfM
implementations, which is reflected in this work, is that SfM can only sense the surface -- whether that it is the ground/snow
surface or the top of a vegetation canopy (Westoby et al., 2012). This makes it fundamentally inappropriate for sub-canopy
mapping of snow. Sub-canopy snow depth mapping with UAV-SfM therefore becomes an exercise in interpolation between
areas of open vegetation rather than sensing the actual snow depth under the canopy. The ability of UAV-lidar to map snow-
depths, with and without canopy cover, with RMSE's <0.17 m is a major improvement on previous attempts.  This RMSE is
comparable to previous efforts with UAV or airborne-SfM and airborne-lidar that have been focussed on mapping the snow
depth of open snow surfaces by masking out forested domains. In applications of airborne-lidar to forested areas much larger
errors have been reported than 0.14 m RMSE (Deems et al., 2013).

### 4.2 Bare surface point cloud coverage is critical

The point coverage of UAV-lidar is the main advantage over UAV-SfM when trying to map sub-canopy snow depth. While
snow depth accuracy at times can be similar between techniques, the ability of UAV-lidar to sense a surface below vegetation
is critical to develop a coherent snow surface DSM. An example of a point cloud cross-section of a UAV-SfM and UAV-lidar
in Figure 7 emphasizes this point. The UAV-SfM data will have wider gaps in the point cloud beneath individual trees that
require interpolation. Features such as tree wells, where the snow depth decreases with proximity to a tree due to
interception/sublimation losses and radiative melting (Pomeroy and Gray, 1995; Musselman and Pomeroy, 2017), will be
missed. An interesting dynamic of the RMSE errors is that while lidar is comparable across all the sites and vegetation
categories, the UAV-SfM RMSE values are much greater in the mountain domain. This is attributed to interpolation artifacts.
In the Prairies where topography is fairly flat, interpolation of the few gaps can give a reasonable approximation of the actual
surfaces. In contrast mountains have much more complex topography and the interpolation of large gaps misses much of the
small scale topography and snow-vegetation interaction features. Interpolation works better between two points that are on the
same plane (prairies) rather than on a complex non-linear slope (mountains) and where gaps in the point cloud are smaller.



### 4.3 Lidar snow depth maps and quantifying snow-vegetation interactions


The ability of UAV-lidar to map sub-canopy snow depth is established by the consistent error metrics reported as well as the continuous bare surface point cloud coverage. The dynamics of snow depth at snow-vegetation process-resolving scales can therefore be examined. Two examples are presented here to foreshadow some of the analyses available with UAV-lidar.

#### 4.3.1 Fortress Snow Depth Change.


The differences between open and forest snow cover processes can be resolved by considering the difference in snow depth between UAV-lidar scans that took place February 13 and April 25, 2019 at Fortress. Over this interval there was intermittent precipitation totaling approximately 100 mm. Measured change in snow depth visualizes how snow-vegetation interactions translated this snowfall into a snow depth distribution change over a two month interval (Figure 10). The upper, open terrain clearly demonstrates the influence of blowing snow redistribution. In the Figure 10c transect there was accumulation of up to


2 m over the period on lee slopes, whilst the upper windswept portions of the ridge demonstrate snow erosion. The dynamics and extents of blowing snow sources and sinks are clearly visualized, as similarly noted by Schirmer and Pomeroy (2019) using SfM. Considering the forest slopes brings out features that UAV-SfM cannot observe. The UAV-lidar can observe the increasing snow drifts in the tree line (in the krummholz and tree islands – blue areas on top of facing slope in Figure 10a). Within the forested (Figure 10b) transect, there is a general decline in snow depth with variability due to melt on a south facing


slope (on left of figure), and development of a tree well in the middle of the transect. The Figure 10b transect demonstrates the lack of wind redistribution in the canopies relative to the Figure 10c transect on the ridgeline.







Figure 10: a) UAV-lidar derived snow depth difference between Feb 13 and Apr 25, 2019. Green and yellow lines in a) correspond to the forest and ridge line transect locations for cross-sections in b) and c) respectively. Cross-section figures plot the 0.5m wide point cloud cross section from the September 19, 2018 snow-free scan (black points) to show the point cloud and the processed surfaces of the bare ground from September 19, 2018 (red), and snow surface from February 14, 2019 (green) and April 25, 2019 (blue) UAV-lidar scans.





### 4.3.2 Prairie peak snow peak depth and ablation patterns

In the prairies, wind redistribution is the main driver of snow depth spatial variability. Areas of tall vegetation accumulate
wind-blown snow from large upwind sources areas and so are typically associated with the deepest snowpacks. In the winter
of 2019, the chronology of snow, temperature, and wind events defined the final snow depth distribution (Figure 11a). The
UAV-lidar flown on March 13 captures all of these interactions. Deep snow drifts are found in the roadside ditches (linear
features of 1.5m snow depth on the north and north west corners Figure 11a), in the edges of wetland vegetation (>1m snow
depths on edges of wetlands identified by green polygons in Figure 11a), and the development of a sastrugi dune complex in
open areas (parabolic dune shapes and small scale snow depth variability in middle of Figure 11a). Areas that the UAV-lidar
was able measure correspond to areas where snow depth are the deepest and have important snow-vegetation interactions. In
contrast UAV-SfM struggles with sensing snow depth in the short shrubs on the edges of wetlands. In the prairies, mapping of
the areas with deep snow is critical as the deepest snow areas are the ones that dominate runoff generation and runoff
contributing area, are critical for ephemeral wetland ecology, and have the longest snowcover duration with the related runoff
timing implications (Fang and Pomeroy, 2009; Pomeroy et al., 2014).



**Figure 11: Peak snow depth at the Rosthern site from UAV-lidar scan on March 13, 2019 a) and snow melt depth difference from UAV-lidar scans on March 18 and March 22, 2019 b). Snow surface near infrared (NIR) reflectance c) and snow depth change d) over a transect (green line in b) are plotted with a hex plot (to show variability) and smoothed line (to show mean change). Green polygons in a) highlight wetland areas.**

Prairie snowpacks are shallow, leading Harder et al. (2016) to conclude that UAV-SfM was unable to capture snow ablation patterns as the signal to noise ratio in the open domain was too large and vegetated area errors were not considered. With the demonstrated ability of UAV-lidar to consistently map shallow snow in open areas and deep snows in the vegetated areas this can be reattempted. Consider the difference in snow depth between March 18 and 23 (Figure 11b) which represents the earliest part of the active melt period in this particular snowmelt season. Two examples of the spatial variability of process interactions can now be visualized at the appropriate resolutions. First, the spatial variability of albedo is a major driver of snowmelt. The greatest melt occurs alongside the gravel-covered "grid" roads in the ditches where road dust significantly lowers the albedo





thereby accelerating melt of the deep snowpacks. Moving eastward from the road ditches into the open fields there is a decrease

in snowmelt depth in the overall scene, visualized in the Figure 11d transect. This pattern is due to the redistribution of dust from the grid roads to the open field snow surface by the prevailing westerly winds. A snow surface dust concentration gradient develops over the winter with higher concentrations of dust in the west than the east. Near-infrared (NIR) reflectance data, a proxy for snow surface albedo from a coincident multispectral UAV flight part of a separate study and not discussed further, demonstrates an increase in albedo (Figure 11c). This increase in albedo corresponds to a decrease in snowmelt rate (Figure

11d), easterly away from the grid road. The gradient in dust and albedo describes the increases in snowmelt rates downwind of the grid road. Second, the spatial variability of snowpack cold content influences melt rates in the early part of the melt season. Within the agricultural field, the sastrugi drifts are not melting – due to the larger cold content of the deep cold snowdrifts relative to the smaller cold content of the shallower surrounding snowpacks. This is also prevalent in the non-melting deep snowdrifts at the vegetated wetland edges. With UAV-lidar, a complete picture of the early and asynchronous

snowmelt processes is possible. If reliant on UAV-SfM the interpolation needed to fill gaps in the point cloud, near vegetation and tops of the sastrugi, will obscure the full spatial pattern of snow depth change that conveys the heterogeneity of ablation processes. The high spatial resolution and vertical accuracy of UAV-lidar is required to capture these spatial patterns as the length scales of the snow surfaces features of interest are small, i.e. sastrugi drifts are on metre scales, and their changes at daily timesteps are at the centimetre scale.

The processes visualized in the Fortress and Rosthern examples are not new, but the value of UAV-lidar is that spatial patterns and changes can be observed across complex landscapes and vegetation gradients with a consistent resolution and accuracy. UAV-lidar will therefore be a powerful tool to understand the landscape scale snow-vegetation interactions as well as make a core contribution to the validation and improvement of distributed modelling of snow processes.

### 4.4 Are the costs and logistics of UAV-lidar worth it?

UAV-lidar, relative to UAV-SfM, provides a superior observation of snow depth below vegetation canopies but it does come at a higher cost and logistical complexity. There are many similarities between approaches and one commonality is that both UAV-lidar and UAV-SfM require access to a GNSS solution to geolocate point clouds in absolute space. The Leica GS16 package used here is on the expensive side of the spectrum ($70,000 CAD) and cheaper products, subscription to virtual reference station networks if available in the study area, or equipment rentals are all viable alternatives to lower costs. The

main cost difference is therefore in terms of the sensor type. A plethora of UAV-SfM options with and without RTK or PPK photo geotagging are available and can range from small inexpensive systems like a consumer grade UAVs (DJI Phantom 3 < $2,000 CAD) or more expensive options like the Sensefly EbeeX PPK system ($30,000 CAD) used here. Current integrated lidar systems suited to snow mapping (laser wavelengths < 1000 nm, small size, weight, and power requirements, and absolute errors < 5 cm) remain an order of magnitude more expensive than UAV-SfM. The cost of the complete UAV-lidar system

(lidar, IMU, software suite, and UAV) used here approached $300,000 CAD. New and cheaper UAV-lidar sensor options are coming to market all the time, largely driven by the sensing advances coming from development of autonomous vehicles, but





these need testing and still require high grade IMU/GNSS solutions to allow for absolute geolocation of point clouds. An underappreciated aspect of UAV-lidar is that the IMU/GNSS solutions can often be more expensive than lidar sensor itself. The additional cost of UAV-lidar to increase sub-canopy snow depth accuracy in dense forest situations in this application can

be simplified to $15,000 CAD per cm reduction in RMSE (difference in equipment costs/difference in Fortress-Tree RMSE). Logistical differences between UAV-lidar and UAV-SfM are more subtle than the stark cost difference. UAV-SfM simply requires a UAV platform and camera in its basic configuration and therefore high endurance, small platforms, with small batteries can be easily deployed to map large areas. In contrast UAV-lidar needs larger platforms that require more cycles of large battery sets to cover similar areas which represents a logical challenge in cold and remote areas. Previous UAV-SfM

experience (Harder et al., 2017) demonstrated the need to utilise GCPs even with PPK/RTK photo geotagging to minimise the bias error metric. The low bias of UAV-lidar errors, without assimilating GCPs, removes the need to deploy GCPs for UAV-lidar applications which can be a large time sink. Data processing software suites and workflows are distinct but ultimately the same level of geomatics expertise is needed to generate useable information. Despite the lower cost and simpler logistics the errors and artefacts that UAV-SfM introduce in the sub-canopy domain, as detailed in sections 4.3.1 and 4.3.2, results in the

noise obscuring the signal (Harder et al., 2017) particularly in dense forest situations. If accurate sub-canopy snow depth is required UAV-lidar is the superior option and therefore worth the added logistics and costs.

## 4.5 Ongoing Challenges and Future Research Needs

The ability of UAV-lidar to resolve sub-canopy snow depths is not without challenges. Precise classification of surface points from snow and ground scans is needed to resolve the snow depth at the resolution to confidently capture snow-vegetation

interactions. The accuracy and resolution demands mean that bare surface classification techniques suitable for airborne platforms that efficiently resolve topography and hydrography at watershed scales from last returns will be unsuitable for resolving the snow depth around a particular shrub from a dense point cloud for example. Sub-canopy snow depth mapping requires careful selection of the appropriate point cloud classification and filtering tools and associated parameters to achieve desired quality and precision in a final point cloud. To preserve the small-scale surface variability point cloud processing will

be less efficient as all points need consideration and the focus on small-scale features will at times lead to erroneous inclusion of points representing large scale non-surface objects. The algorithm and parameters decisions also have to be adjusted for each flight and site/environment for UAV-SFM due to the variable quality and noise of the generated point cloud.

An especially challenging feature in resolving a ground surface is the presence of low and dense vegetation such as shrubs and wetland reeds. This is evident in looking in the centre of the wetland zones (green polygons) of Figure 11a where there are

negative snow depths calculated. In this case, the lidar pulses cannot penetrate the dense vegetation to the underlying ground surface and the classified bare ground surface points have a positive bias. As snow accumulates, the reeds compress and shrubs bend over to the extent that the corresponding snow surface is below the biased bare ground surface. In the examples presented above, the areas of negative snow are limited to areas where snow depth is shallow and are not as critical to capture as the deep snow in the wetland edges. This challenge might also be apparent in other regions, such as the Arctic tundra, where shrub





bending and burial by snow has been extensively documented (Pomeroy et al., 2006; Sturm et al., 2005). While shrubs are much sparser than wetland reeds their dynamic change in height and potential to bias positively the ground surface extraction will increase uncertainty of snow depth estimation in hydrologically significant snow accumulation areas. More powerful lasers and higher scan rates may be possible to increase point cloud density and penetration to the ground surface but leads to sensors that may exceed capabilities of most UAV platforms. Advances in bare surface classification software tools to address

the large noise associated with low and dense vegetation is an obvious avenue of improvement. This avenue is inherently limited, as even a perfect bare surface extraction algorithm will not identify points at the ground surface if pulses cannot penetrate to the ground surface. The time of year chosen for the ground surface scan, ideally right after snowmelt when vegetation is at its lowest and not growing yet, may minimize errors. Unfortunately, this may not be feasible if the critical wetland areas are inundated as is often the case in the Canadian Prairies in spring.

Mapping sub-canopy snow depth is important but the ultimate variable of interest is SWE. The challenge is that at snow-vegetation interaction scales there may be significant variability from snow pack densification being driven by different processes across a landscape (Faria et al., 2000). Densification from wind packing is prevalent in open areas versus metamorphic densification due to temperature gradients in sheltered sub canopy areas (López-Moreno et al., 2013). Current methods of modelling or measuring snow density are not without problems at these small scales. Modelling snow density will

impose conceptual understandings of these processes (Painter et al., 2016) which may be inappropriate for the small scale features that need to be represented – these may miss mechanical densification from snow clumps unloading or dripping from the canopy for example.  Observational approaches are also a challenge as typical *in situ* measurements are destructive, limited in extent, and often too limited to develop robust relationships of depth versus density at the small scales needed (Kinar and Pomeroy, 2015a; Pomeroy and Gray, 1995). Opportunities may be available to pair UAV-lidar with other UAV-borne sensors

such as passive gamma ray or snow acoustics (Kinar and Pomeroy, 2015b) to develop higher resolution estimates of snow density.

**5 Conclusions**

Remote sensing techniques to determine snow depth have consistently been challenged by the presence of vegetation. This has complicated efforts to observe and understand snow-vegetation interactions at the necessary spatial scales. This work directly

considers emerging UAV-lidar and UAV-SfM techniques to address this gap in observational capacity. Based upon extensive data collection at a variety of sites and snow conditions with varying snow-vegetation processes, the ability of UAV-lidar to measure sub-canopy snow depth is demonstrated. UAV-lidar provides snow depth estimates with RMSE's <0.1 m in open areas and <0.17 m in vegetated areas. The UAV-lidar metrics consistently exceed the UAV-SfM metrics and are better than previously reported results in the airborne-lidar and UAV-SfM literature. The ability of UAV-SfM to measure snow depth in

open areas is validated with respect to the growing body of literature and reconfirms that UAV-SfM is fundamentally inappropriate to sense sub-canopy surfaces. The clear advantage of UAV-lidar is that, as an active sensor, it provides a high

point cloud density that is unaffected by surface homogeneity and allows for reliable bare surface detection. With UAV-lidar we can now confidently observe sub-canopy snow depth at centimetre scales needed to examine snow-vegetation interactions at research catchment extents (ie <5 km$^2$). UAV-lidar is an emerging tool that will contribute to improving basin-scale snow

accumulation estimates, validation and parametrisation of distributed snow models, and enhancing snow vegetation interaction process understanding over the landscape scale.

## Code/Data Availability

The data underlying this analysis and its documentation is available at https://dx.doi.org/10.20383/101.0193 under a Creative Commons CC-BY-4.0 license. The LAStools workflows and R code used to complete the analysis are available from

https://github.com/phillip-harder/UAV-snowdepth under a GNU General Public License v3.0.

## Author contribution

PH designed the field campaigns, performed the data collection, and completed/managed the data processing and analysis. PH prepared the manuscript with contributions from all authors.

## Competing interests

The authors declare that they have no conflict of interest.

## Acknowledgements

Grateful acknowledgment of field and data processing assistance from Dong Zhao, Alistair Wallace, Greg Galloway, Robin Heavens, Lindsey Langs, Cob Staines, Andre Bertoncini, and Bosse Sottmann. The support of Fortress Mountain Ski Resort, the Natural Sciences and Engineering Research Council of Canada (NSERC), the Canada Research Chairs programme, Canada

First Research Excellence Fund through the Global Water Futures programme, and the Canadian Department of Western Economic Diversification made this study possible.

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
