# Peer review of "Improving sub-canopy snow depth mapping with unmanned aerial vehicles: lidar versus structure from motion techniques"

_The Cryosphere, 2019_

## Referee Comment (RC1) · Anonymous Referee #1 · 19 Feb 2020

Please see attached .doc if not formatted properly

The Cryosphere Manuscript tc-2019-284

Title: Advances in mapping sub-canopy snow depth with unmanned aerial vehicles using structure from motion and lidar techniques Authors: Phillip Harder, John W. Pomeroy, and Warren D. Helgason

Paper Summary:

The authors show a comprehensive comparison between snow depth derived from UAV structure from motion and UAV lidar. They compare both datasets in forested

areas, shrub areas, and in open/smoother terrain to manual snow depth measurements that are geolocated with GNSS systems. The authors show that UAV lidar can provide information beneath the canopy. This allows the user to look at snow depth variability and snow-vegetation processes with lidar. The authors clearly show issues with UAV SfM. The authors also nicely show a cost comparison stating that lidar is more accurate but costs ∼15,000 dollars per additional cm of accuracy. The paper is well written and it discusses many caveats and issues that remain with lidar. The paper is a nice demonstration of the accuracy of UAV lidar, its utility, and remaining limitations.

The authors do not just evaluate the two techniques. The authors show how lidar can capture fine scale variability, such as tree wells, and detect fine scale processes with prairies. This shows originality and significance. I recommend the paper be published pending minor revisions.

General/Major Comments:

No major comments. Mostly, nit-picky comments. Enjoyed the paper, particularly Figure 7 and Figure 10 and their ability to capture tree wells and their changes throughout time.

Specific Comments:

Title sounds like a review paper. Perhaps consider something like, UAV lidar improves observations of sub-canopy snow depth variability over UAV SfM.

Line 7: I would disagree that techniques are lacking. You might say something related to that they don't always exist; satellite remote sensing is difficult. Airborne lidar captures this. So does TLS. This has been shown.

Line 26: Traditional remote sensing methods is vague. What's traditional to you might to be traditional to someone else.

Line 35: I would just say test processes

[Figure]

Line 38: I don't think Painter et al. 2016 initialized or validated a model. Andrew Hedricks recent WRR paper (Hedrick et al., 2018) would be better suited, which uses ASO data to update iSnobal (reinitialize).

Line 66: Leading to variably, I think you mean variability

Line 70: It would be great to reference (Currier et al., 2019) here. Table 1 in their paper reviews this and they provide their own evaluation metrics of ALS in a forest and open area. I would also reference (Mazzotti et al., 2019). They showed a comparison of lidar in Switzerland to snow depth transects in forested areas as well.

Line 75: TLS was used in the forest in (Currier et al., 2019). Yes, the TLS did not go all the way into the entire forest but from an evaluation perspective of airbone lidar or SfM there's little difference from being 300 meters in a forest as long as there are consistent trees overhead that would inhibit returns from the laser. Also, their paper did not explicitly show that TLS couldn't be used further in the forest, it just gets more complicated.

Line 90: Could add that (Zheng et al., 2016) lidar to understand vegetation processes effect on snow. They particularly note bias that might occur due to tree wells. (Currier & Lundquist, 2018) used lidar to understand the snow-vegetation interactions in multiple climates. (Mazzotti et al., 2019) also used airborne lidar data to improve the understanding of snow depth related to the forest in Colorado and Switzerland.

Line 190: I would mention here that the code is provided on your github page. Great job with providing this.

Line 205: Trees typically are taller than 50 cm. Most people consider a tree to be at least 2 m tall. Why did you choose 50 cm? This is inconsistent with what the caption shows in Figure 4.

Line 230: What is estimated and what is observed? I'd say UAV-derived Snow Depth and Snow Depth Probe Manual Observations, or something more specific.

Line 235: Yes, the reported error metrics are inflated when moving into the forest. It'd be worthwhile mentioning that the sample size is much less. Some lidar points do great. In the methods the GNSS mentions a $\pm 2.5$ cm accuracy, how was that determined. Is it possible that this is inflated when in the forest? If not, mention that. Are these errors from how the point cloud was processed and points were classified? Is $\pm 2.5$ cm true for both horizontal and vertical accuracy?

Line 238: I'd start a new paragraph when introducing the error metrics with SfM.

Line 245: The authors should be using Digital Terrain Models instead of Digital Surface Models throughout.

Figure 6: Cool analysis. I would consider adding a black dashed line for 2.5 cm. This plot supports the results of Currier et al. 2019, that the airborne lidar is more likely to penetrate the shrubs than the TLS observations. What's the scientific name for the shrubs found at these locations?

Figures: I would change the easting northing to the total number of meters within the domain, or start at 0 and show ticks from 0 m. I don't know the projection information, and if I did the numbers aren't that meaningful. If the location is important, please provide the UTM zone. But still it's a bit annoying to do the subtraction each time to get a sense of scale. I would just make it easier for the readers, if possible. Otherwise the figures are great.

Line 317: This seems like an appropriate time to re-mention UAV lidars ability to capture tree wells.

Line 321: Confusing sentence. Deems reported errors in the forest larger than 14 cm? Why is 14 cm mentioned. Figure 5 reports RMSE of 0.15 and 0.16.

Also, in the previous sentence. Studies have masked out the forest? Studies have looked at airborne lidar accuracy in the forest.

Line 355: Really cool figure and analysis

Line 375: Green polygons look cyan when zoomed out, might choose a different color. Furthermore, the near infrared data seemingly comes out of nowhere – maybe provide some more context within the section for it and why it needs to be mentioned. Provide a citation for NIR serving as a proxy for albedo. Line 435: "The accuracy and resolution demands mean that bare surface classification techniques suitable for airborne platforms that efficiently resolve topography and hydrography at watershed scales from last returns will be unsuitable for resolving the snow depth around a particular shrub from a dense point cloud for example" The paper did not show that using the last returns was unsuitable. The classification technique used something similar to last returns. Previous studies have showed using the last returns resulted in a generally unbiased snow depth estimate, and provided a reasonable approximation of the variability. I am not sure what this sentence is attempting to say. Line 465: A discussion referencing the difficulties with modeling in Mark Raleigh's paper seems appropriate and a better citation then Tom Painters 2016 paper. Furthermore, when mentioning snow pack density variability, mentioning Karl Wetlaufer's paper seems appropriate (Raleigh & Small, 2017; Wetlaufer et al., 2016). Line 479: "The UAV-lidar metrics consistently exceed the UAV-SfM metrics and are better than previously reported results in the airborne-lidar and UAV-SfM literature." This isn't true. Metrics are similar but not better than. Please note line 69.

References Currier, W. R., & Lundquist, J. D. (2018). Snow Depth Variability at the Forest Edge in Multiple Climates in the Western United States. Water Resources Research, 54, 1–18. https://doi.org/10.1029/2018WR022553 Currier, W. R., Pflug, J., Mazzotti, G., Jonas, T., Deems, J. S., Bormann, K. J., et al. (2019). Comparing aerial lidar observations with terrestrial lidar and snow‑probe transects from NASA's 2017 SnowEx campaign. Water Resources Research, 1–10. https://doi.org/10.1029/2018wr024533 Hedrick, A. R., Marks, D., Havens, S., Robertson, M., Johnson, M., Sandusky, M., et al. (2018). Direct Insertion of NASA Airborne Snow Observatory-Derived Snow Depth Time Series Into the iSnobal Energy Balance Snow Model. Water Resources Research, 54, 8045–8063.

https://doi.org/10.1029/2018WR023400 Mazzotti, G., Currier, W. R., Deems, J. S.,
Pflug, J. M., Lundquist, J. D., & Jonas, T. (2019). Revisiting Snow Cover Variability and
Canopy Structure Within Forest Stands: Insights From Airborne Lidar Data. Water Re-
sources Research, 55(7), 6198–6216. https://doi.org/10.1029/2019wr024898 Raleigh,
M. S., & Small, E. E. (2017). Snowpack density modeling is the primary source of
uncertainty when mapping basin-wide SWE with lidar. Geophysical Research Letters,
44(8), 3700–3709. https://doi.org/10.1002/2016GL071999 Wetlaufer, K., Hendrikx,
J., & Marshall, L. (2016). Spatial heterogeneity of snow density and its influence on
snow water equivalence estimates in a large mountainous basin. Hydrology, 3(1).
https://doi.org/10.3390/hydrology3010003 Zheng, Z., Kirchner, P. B., & Bales, R. C.
(2016). Topographic and vegetation effects on snow accumulation in the southern
Sierra Nevada: A statistical summary from lidar data. Cryosphere, 10(1), 257–269.
https://doi.org/10.5194/tc-10-257-2016

Please also note the supplement to this comment:
https://www.the-cryosphere-discuss.net/tc-2019-284/tc-2019-284-RC1-
supplement.pdf
* * *
**TCD**

Interactive
comment

---

## Referee Comment (RC2) · Anonymous Referee #2 · 30 Mar 2020

Paper Summary:

The authors compare two relatively new methodologies for using UAVs for mapping snow depths in forested and open prairie environments with in situ ground validation GNSS surveys. They present a very thorough analysis involving an impressive collection of data from 19 unique survey dates from two distinct environments over the course of a single winter season. The time and effort taken to plan, collect, and process such a comprehensive dataset cannot be overstated! The results of the comparison on the ability of both the UAV-lidar and UAV-SfM to estimate snow depths are not necessarily new, but to my knowledge, they have not been compared as exten-

sively with both the successes and failures of both methodologies clearly presented. In open environments, the UAV-lidar and UAV-SfM snow depth mapping capabilities are similar, but in vegetated areas, the UAV-lidar methods excel by having the ability to penetrate through vegetation and measure sub-canopy snow depth. However, in densely vegetated, tight canopy environments, even the UAV-lidar mapping method cannot penetrate the canopy and therefore cannot produce reliable snow depth estimates. An added benefit of using the UAV-lidar over UAV-SfM for snow depth mapping is the insensitivity of the lidar to homogeneous surface conditions and variable/poor solar illumination, both of which contribute to substantial errors in UAV-SfM mapping. In-addition, the increased vertical accuracy of the UAV-lidar sensors can be used to better detect patterns in snow distribution and depth previously not obtainable over basin-wide study sites in complex landscapes. The authors do a nice job at presenting their findings in a well-written manner using suitable figures. As an added bonus, the authors also discuss the cost difference between the UAV measuring methodologies, and calculate a metric that assigns a dollar value to each centimeter of improved RMSE between methods. This cost analysis is of interest, but probably has less relevance for the future, as the price for the type of equipment used in this study continues to decrease dramatically year-by-year. I recommend the publication of this paper pending minor revisions addressing the suggested comments and technical edits.

A PDF supplement has also been uploaded that contains all the suggested edits/comments. In the technical edits section, this PDF supplement has all changes highlighted in BOLD.

An example of the suggested changes to Figure 7a has also been uploaded as Figure 1 – Slide 1.JPG. This example figure provides a visualization of the changes being suggested for Figures 7a, 8a, 9a (applies to General Comment at Line 270/295/300).

General Comments:

Line 59 – 'differencing snow-covered (hereafter snow) and snow-free (hereafter

ground). . .' Double check terminology throughout paper for consistency. The following different term are used: bare-ground, bare ground, ground, surface, bare surface. Personally – I like the use of the term bare-ground.

Line 59 – 'Digital Surface Models (DSMs)' I think you are actually referring to the Digital Terrain Models (DTMs). Change this reference throughout the paper.

Line 134 – 'flight parameters to maximise mapping efficiency were set to. . .' What about limiting the scan angle? The Riegl lidar can scan 360 degrees, what level of off nadir scan angle did you limit the data collection/processing to and why?

Line 135 – '100 m flight altitude above the surface. . .' Did the mission planning software make use of terrain following mode to ensure consistent flight altitude above ground? If so, what source of terrain information did you use?

Line 148 – I deleted the term differential: differential GNSS corrections (code-based) are significantly less, accurate than RTK/PPK/PPP (carrier phase methods) – I suspect even though the Leica GS16 unit is DGPS capable, you used the more accurate carrier phase correction methods.

Line 150 - suggest removing the term 'random within the survey areas and' if the transects were also selected to most efficiently survey the greatest variety of vegetation types.

Line 152 – 'provided a real-time-kinematic (RTK) survey solution . . .' While conducting your manual surveys did you make use of the RTK capabilities – or did you post-process the rover data as indicated at line 153?

Line 152 – 'accuracy of $< \pm 2.5$cm.' Can you provide a reference for this?

Line 154 – '(https://webapp.geod.nrcan.gc.ca/geod/tools-outils/ppp.php)' Add this website to the references section

Line 154 – 'absolute base station location.' How long did you collect your raw GNSS

data for and what were the PPP computed standard deviations for the base station locations? Did you always use the same base station location for every flight?

Line 174 – '<2.5 cm.' Do you mean +/- 2.5 cm as mentioned earlier in the text? Is this value based on the specs of the Leica GS16 GNSS survey equipment or was it based on the PPP online standard deviations? How did you obtain this value?

Line 181 – '< ±2.5 cm' Same comment as above? Is this value based on the specs of the Leica GS16 GNSS survey equipment or was it based on the PPP online standard deviations? How did you obtain this value?

Line 205 – 'vegetation height (open <0.1 m, shrub <0.5 m, and trees >0.5 m)...' These values differ from what is in the Figure 4 caption. Which vegetation height classes did you use, and how did you choose the class heights?

Line 223 – I deleted reference to RTK - In line 55 you indicate the rover survey points were post-processed, therefore I am assuming you used a PPK GNSS solution here?

Line 230 – 'points extracted from the point clouds or interpolated surfaces...' This sentence is confusing. It is unclear whether you extracted the UAV snow depth values from the point clouds or the interpolated DSMs? Which one was it?

Line 256 – Figure 6 - Please add to the caption a description of which metrics are visualized by the whiskers of the boxplots.

Line 266 – 'The noisy UAV-SFM points in the middle of the slope challenge the snow surface extraction even without the presence of vegetation leading to an underestimation of the snow surface.' Do you have any idea on why the SfM product detected something in the open areas on the slope? Why does it lead to an underestimation of snow in this area? Based on the Figure 7a cross-section it looks like the UAV-SfM red points are equal to or above the green lidar points. Why did the interpolation go so low? Did the interpolation treat missing points as 0 or bare ground values?

Line 270/295/300 – Figure 7-8-9 - Suggest using shaded/transparent colour bars on

plot a) to indicate the extent of the tree features. This will help highlight the tree well extent and how the UAV-SfM interpolation result in deeper snow values across these features (I have uploaded an example Figure of 7a. that illustrates what I am trying to describe – Slide 1.JPG). Suggest using a more obvious colour in Figure b) for highlighting the SfM only classes. Suggest trying to match the tone of colours in Figure c) to more closely match that used in Figure b). Making the open areas a little bluer, and again highlighting the SfM only points in a more obvious colour. Figure 7b It sort of looks like the SfM only class occur near the edges of the study area in a just a couple areas. Is this related to steeper scan angles at the edge of the study site, perhaps coupled with steep terrain? Figure 9c) I suggest mentioning in the figure caption that the large dark areas of no lidar points represent the extent of the melt water ponds.

Line 288 – the negative UAV-SfM snow depth estimates discussed here are explained at lines 443-450. Perhaps also providing further explanation here might be helpful.

Line 316 – In the example of 7a, the interpolation resulted in erroneously deep snow depth estimates. This will not always be the case and in some instances can result in underestimations depending on the season, elevation, forest type, etc. Many studies have highlighted the differences in snow depths/characteristics between open/forested sites that will influence these interpolation errors. I think providing some further explanation on the type/magnitude of interpolation errors that may occur when using UAV-SfM techniques would help strengthen your findings/statement here.

Line 318 – 'major improvement on previous attempts.' Can you provide some context on what is considered a major improvement, including references to previous studies/RMSEs?

Line 318 – 'previous efforts. . .' Can you provide some references?

Line 321 – '0.14 m RMSE (Deems et al., 2013).' Can you provide the actual magnitude of errors previously reported for comparison in the Deem et al., 2013? What is the significance of this 0.14 m RMSE?

Line 342 – 'intermittent precipitation totaling approximately 100 mm' How was this determined/measured? What kind of uncertainties are associated with this reported precipitation value. I also want to confirm that you mean 10 cm of snow? This seems low for mountain snow.

Line 350 – 'and development of a tree well in the middle of the transect. The Figure 10b transect demonstrates the lack of wind redistribution in the canopies relative to the Figure 10c transect on the ridgeline.' It is unclear where the development of the tree well is highlighted/visible in Figure 10b. It also unclear how Figure 10b demonstrates the lack of wind re-distribution in the canopies. Please provide more detail here.

Line 366 – 'In contrast UAV-SfM struggled with sensing snow depths in the short shrubs on the edges of wetlands.' This sentence contradicts the results displayed in Figure 5, which illustrated that the UAV-SfM had lower RMSE in the shrub class compared to the UAV-lidar. It also does not support the discussion starting at Line 286 and expanded at Lines 443-450, which discusses the challenges that BOTH lidar and SfM face in trying to measure below the canopy in dense shrub vegetation.

Line 467 – 'Observational approaches are also a challenge as typical in situ measurements are destructive, limited in extent, and often too limited to develop robust relationships of depth versus density at the small scales needed (Kinar and Pomeroy, 2015a; Pomeroy and Gray, 1995).' The methods developed by Proksch et al., 2015 do provide a method for measuring snow density at a much smaller scale applicable for these process-scale studies. The Proksch et al., 2015 methods have been recently rigorously applied to a set of snow on sea ice measurements by King et al., 2020, highlighting the ability to document the local-scale variations in snow density relatively quickly over larger spatial extents.

Proksch, M., Löwe, H. and Schneebeli, M., 2015. Density, specific surface area, and correlation length of snow measured by high‐resolution penetrometry. Journal of Geophysical Research: Earth Surface, 120(2), pp.346-362.

King, J., Howell, S., Brady, M., Toose, P., Derksen, C., Haas, C., and Beckers, J.: Local-scale variability of snow density on Arctic sea ice, The Cryosphere Discuss., https://doi.org/10.5194/tc-2019-305, in review, 2020.

Line 474 – 'necessary spatial scales' – Please be more specific on what scales you are referring to.

Technical Comments:

Line 13 – suggest changing to 'measure returns from a wide range of scan angles, increasing the likelihood of successfully. . .'

Line 51 – suggest changing to 'are valuable automated data sources, but are spatially limited in extent and can often suffer from location/elevation bias. . .'

Line 53 – suggest changing to 'and so may not be suitable for snow hydrology calculations or model validations in forested regions even though they are often. . .'

Line 60 – spelling correction: quality

Line 62 – suggest changing to 'pulse can be observed with returns possible from within the canopy and from the sub-canopy ground surface. In contrast UAV-SfM. . .'

Line 64 – spelling correction: variability

Line 80 – spelling correction: focused

Line 87 – punctuation: 'In dense forests, vegetation. . .'

Line 90 – suggest changing to 'increase in snow accumulation over aerodynamically rough surfaces or in sheltered areas where the wind speeds decrease and snow is deposited – this includes forest edges. . .'

Line 98 – suggest changing to 'varies across complex vegetated landscapes. . .'

Line 105 – suggest changing to 'ability of the UAV-lidar and UAV-SfM techniques for measuring snow depth in open
Line 106 - (50.833 N, 115.220 W)

Line 108 – spelling correction: focused

Line 109 – suggest changing to '(Figure 1a – background center)...'

Line 111 – suggest changing to 'alpine ski resort in the 1960's, but is currently a limited-use...'

Line 114 – suggest changing to 'Canadian Prairies were examined in this study.'

Line 117 – correction: remove negative sign if using 'W' to indicate west (51.941 N, 106.379 W) & (52.694 N, 106.461 W)

Line 125 – Figure 1 caption: suggest changing to 'Figure 1: a) Fortress Mountain Snow Observatory in Kananaskis, Alberta Canada, b) Rosthern and c) Clavet prairie study locations in Saskatchewan Canada. Data collection was on Fortress Ridge (background center) an area of high topographic variability and a mix of dense forests and clearings. The Clavet photo highlights the transition zone between the open upland terrain and the lower elevation vegetated wetland. The Rosthern scene highlights the low vertical relief of upland areas and isolated woodlands amongst cultivated fields.

Line 155 – suggest changing to 'GS16 rover points to correct for the PPP updated base station locations were completed using the Leica Infinity software...'

Line 158 – 'suggest changing to 'To assess the accuracy of the UAV snow depth measuring methods, as well as provide insight into the seasonally evolving snow depth/distribution, a total of 19 flight/manual surveys were conducted between all three study sites between September 2018 to April 2019. These are summarised by date, surveyed surface, UAV data collected, and corresponding number of manually surveyed surface elevation points in Table 1.

Line 165 – suggest changing to 'difference between a bare ground DSM and a snow surface DSM.'

Line 176 – suggest changing to 'Finally, overlapping scan data from adjacent flight lines are used to optimise the IMU trajectory, to align the scan lines and reduce the noise of the final point cloud within the RiPrecision tool. This final step in noise reduction can improve the final product because the 1.5 cm laser data precision is greater than the post processed IMU trajectory accuracy. (I used the 15mm stated precision of the Reigl sensor presented earlier in the text to get the 1.5cm value here)

Line 193 – suggest changing to 'For the bare-ground lidar scans, the height of vegetation...'

Line 207 – spelling correction: include

Line 214 – suggest changing to '2.3.6 Point Cloud Density'

Line 221 – suggest changing to '3.1 Accuracy of UAV-lidar versus UAV-SfM snow depth estimates

Line 231 – suggest changing to 'Plots are segmented for points extracted from the point clouds or interpolated surfaces within each vegetation class (rows), sites (columns) and observation method (colours).' – See general comments above about clearing up the confusion concerning which product the points were extract from.

Line 232 – suggest changing to 'The influence of vegetation on estimating snow depths from UAVs can be directly assessed by...'

Line 234 – suggest changing to 'Open Prairie and open Fortress RMSE values are similar (0.09 m and 0.1 m RMSE respectively)...'

Line 235 – suggest changing to 'equally successful at penetrating the open leaf-off deciduous tree canopy at the prairie sites as the closed needleleaf canopy at the Fortress site based on the similar RMSE values within each site's tree vegetation class.'

Line 238 – suggest changing to 'The Open vegetation has a large RMSE range between sites (0.1 m in Prairie and 0.3 m in Fortress respectively) while vegetation class

RMSEs range from. . .'

Line 240 – suggest changing to 'UAV-lidar in the prairie Shrub case, the difference between these techniques is only 0.04 m, which is within the +/- 2.5 cm observational uncertainty of the GNSS survey equipment used in this project.

Line 247 - suggest changing to 'manual GNSS surveys using boxplots (Figure 6). The boxplots in Figure 6 illustrate that the UAV-SfM snow surface elevations. . .'

Line 257 – suggest changing to '3.2 Point cloud density'

Line 263 – suggest changing to 'could not reliably return surface points with a density > 1 pt 0.25 m-2 whilst. . .'

Line 263 – punctuation: 'At Fortress, UAV-lidar. . .'

Line 265 – suggest changing to 'lack of UAV-SfM sub-canopy points identified within the treed vegetation class results in an interpolated snow surface that is erroneously deep under trees, completely missing the detection of the reduced snow depths which are clearly detected (green line) around the base of the trees by the UAV-lidar.'

Line 274 – suggest changing to 'c) with the same overlain transparent point type classification colour scheme as shown in b).'

Line 276 – suggest changing to 'The predominantly open nature of the Prairie sites demonstrates a minimal difference in point density between UAV-lidar and UAV SfM measurement techniques. The average extent of the study domain covered with a point density of > 1 pt 0.25 m2 for 5 coincident flights at the Prairie sites was computed, resulting in the mean coverage of 92% versus 83% of the study area for the UAV-lidar and UAV-SfM respectively.

Line 281 – suggest changing to 'These gaps in the UAV-SfM point clouds are interpolated and therefore will represent. . .'

Line 287 – suggest changing to 'both lidar pulses and SfM solutions interpret the vegetation surface as the top of the bare-ground or snow surface and therefore little difference exists between these two DSMs during all measurement periods. An additional challenge of using the UAV-SfM techniques is that large gaps in points appear beneath the tall wetland edge vegetation due to the inability to penetrate the sub-canopy, as visualized in the cross-sections of Figure 8a and 9a, where the estimated UAV-SfM snow surface is below the UAV-lidar ground surface.'

Line 316 – suggest changing to 'Sub-canopy snow depth mapping with UAV-SfM therefore becomes an exercise in interpolating snow depth values observed in open areas without vegetation to areas with dense vegetation, rather than sensing the actual snow depth under the canopy.'

Line 322 – suggest changing to '4.2 Bare-ground point cloud density is critical'

Line 323 – suggest changing to 'The increased point density of UAV-lidar. . .'

Line 325 – suggest changing to 'The point cloud cross-sections illustrated in Figure 7 emphasize these findings, highlighting the wider gaps in the UAV-SfM point cloud beneath individual trees that require interpolation over longer distances resulting in greater potential for error.' (The lidar data also requires interpolation)

Line 332 – suggest changing to 'In contrast, mountainous regions have much more complex topography. . .'

Line 337 – suggest changing to 'continuous bare-ground point cloud coverage.'

Line 338 – suggest difference word choice for: foreshadow

Line 340 – suggest changing to 'Differences between open and forest snow cover processes can be explored by examining the difference in snow depth. . .'

Line 342 – suggesting changing to 'UAV-lidar measured change in snow depth visualizes. . .'

Line 343 – suggest deleting line: 'The upper, open terrain clearly demonstrates the

influence of blowing snow redistribution' because this sentence is ambiguous.

Line 343 – suggest changing to 'In the Figure 10c transect cross-section there was accumulation of up to 2 m over the September-April time period on lee slopes, whilst the upper windswept portions of the ridge demonstrate snow erosion between February and April."

Line 346 – suggest changing to 'The dynamics and extents of blowing snow sources (grey/red) and sinks (blue) are clearly visualized in 10a, which closely match the findings of Schirmer and Pomeroy (2019) using SfM for this same study region.

Line 347 – suggest deleting line: 'Considering the forest slope brings out features that UAV-SfM cannot observe.' Because this sentence appears as a fragment

Line 349 – suggest changing to 'there is a general decline in snow depth from February to April (due to melt on the south facing slope).'

Line 360 – suggest changing to 'wind-blown snow from open upwind sources and are typically associated with. . .'

Line 366 – suggest changing to 'Areas that the UAV-lidar was able to measure correspond to areas. . .'

Line 390 – suggest changing to 'This gradient in dust and albedo is likely associated with the increases in snowmelt rates observed downwind of the grid road.'

Line 405 – suggest changing to 'UAV-lidar, relative to UAV-SfM, provides the ability to measure snow depth below vegetation. . .'

Line 408 – suggest changing to 'and cheaper equipment, subscriptions to virtual reference station networks if available in the study area (requires only a rover and not a base station), or equipment rentals are all viable alternatives to lower costs.'

Line 410 – suggest changing to 'The main cost difference between UAV-lidar and UAV-SfM platforms is therefore in terms of the UAV sensor payload.'

Line 412 – suggest changing to 'like consumer grade UAVs (DJI Phantom 3 < $2,000 CAD), to more expensive options like...'

Line 413 – suggest changing to 'Current integrated lidar systems suited to UAV snow mapping'

Line 423 – suggest changing to 'In contrast, most current UAV-lidar configurations need larger platforms that require more cycles of large battery sets to cover similar areas, which represents a logistical challenge in keeping the batteries warm and charged in cold and remote areas.'

Line 428 – suggest changing to 'Despite the lower initial purchase cost and longer flight endurance, the errors and artefacts that UAV-SfM measuring techniques introduce in sub-canopy snow depth measurements, as detailed in sections 4.3.1 and 4.3.2, suggest that UAV-SfM is not able to directly measure snow depth in densely vegetated environments.'

Line 434 – suggest changing to 'Precise classification of surface points from snow and ground scans are needed to resolve...'

Line 435 – suggest changing to 'The accuracy and resolution demands are such that bare-ground surface classification techniques developed for airborne platforms to resolve topography and hydrography at watershed scales from lidar last returns may be unsuitable for resolving snow depths.'

Line 438 – suggest changing to 'filtering tools and associated parameters to be able to reliably detect the sub-canopy bare-ground surface and achieve desired quality...'

Line 441 – spelling correction: 'large-scale'

Line 448 – suggest changing to 'the areas of negative snow are limited to areas where snow depth is relatively shallow in comparison to the deep snow in the wetland edges.'

Line 452 – suggest changing to 'snow depth estimation in these hydrologically significant snow accumulation areas.'

Line 453 – suggest changing to 'ground surface, but current sensors with these characteristics may exceed the payload capacities of most UAV platforms. Advances in bare surface classification/filtering software...'

Please also note the supplement to this comment:
https://www.the-cryosphere-discuss.net/tc-2019-284/tc-2019-284-RC2-supplement.pdf
* * *
[Figure]

**Fig. 1.** Example of suggested change to Figure 7a.

---

## Author Comment (AC1) · 28 Apr 2020

**The Cryosphere**
**Manuscript tc-2019-284**
**Title:** Advances in mapping sub-canopy snow depth with unmanned aerial vehicles using
structure from motion and lidar techniques
**Authors:** Phillip Harder, John W. Pomeroy, and Warren D. Helgason
**Paper Summary:**
The authors show a comprehensive comparison between snow depth derived from UAV structure from motion and UAV lidar. They compare both datasets in forested areas, shrub areas, and in open/smoother terrain to manual snow depth measurements that are geolocated with GNSS systems. The authors show that UAV lidar can provide information beneath the canopy. This allows the user to look at snow depth variability and snow-vegetation processes with lidar. The authors clearly show issues with UAV SfM. The authors also nicely show a cost comparison stating that lidar is more accurate but costs ~15,000 dollars per additional cm of accuracy. The paper is well written and it discusses many caveats and issues that remain with lidar. The paper is a nice demonstration of the accuracy of UAV lidar, its utility, and remaining limitations.
The authors do not just evaluate the two techniques. The authors show how lidar can capture fine scale variability, such as tree wells, and detect fine scale processes with prairies. This shows originality and significance. I recommend the paper be published pending minor revisions.

**General/Major Comments:**
No major comments. Mostly, nit-picky comments. Enjoyed the paper, particularly Figure 7 and Figure 10 and their ability to capture tree wells and their changes throughout time.

Thank you for the detailed review. The edits you suggest will make this a much stronger contribution- see the specific responses in red below.

**Specific Comments:**
Title sounds like a review paper. Perhaps consider something like, UAV lidar improves observations of sub-canopy snow depth variability over UAV SfM.

Good point- this is definitely not supposed to be a review paper.  Changed to "Improving sub-canopy snow depth mapping with unmanned aerial vehicles: lidar versus structure from motion techniques "

**Line 7:** I would disagree that techniques are lacking. You might say something related to that they don't always exist; satellite remote sensing is difficult. Airborne lidar captures this. So does TLS. This has been shown.

Agreed- changed to "Vegetation has a tremendous influence on snow processes and snowpack dynamics yet remote sensing techniques to resolve the spatial variability of sub-canopy snow depth are not always available and are difficult from space-based platforms"

**Line 26:** Traditional remote sensing methods is vague. What's traditional to you might to be traditional to someone else.

Traditional = satellite in my mind. Has been changed. "Unfortunately, satellite remote sensing methods…"

**Line 35:** I would just say test processes

Changed

**Line 38:** I don't think Painter et al. 2016 initialized or validated a model. Andrew Hedricks recent WRR paper (Hedrick et al., 2018) would be better suited, which uses ASO data to update iSnobal (reinitialize).

Agreed and have changed reference.

**Line 66:** Leading to variably, I think you mean variability

Corrected

**Line 70:** It would be great to reference (Currier et al., 2019) here. Table 1 in their paper reviews this and they provide their own evaluation metrics of ALS in a forest and open area. I would also reference (Mazzotti et al., 2019). They showed a comparison of lidar in Switzerland to snow depth transects in forested areas as well.

Have added these references.

**Line 75:** TLS was used in the forest in (Currier et al., 2019). Yes, the TLS did not go all the way into the entire forest but from an evaluation perspective of airbone lidar or SfM there's little difference from being 300 meters in a forest as long as there are consistent trees overhead that would inhibit returns from the laser. Also, their paper did not explicitly show that TLS couldn't be used further in the forest, it just gets more complicated.

We are seeking a tool that can measure snow-depth below forest canopies to further process understandings at the landscape scale rather than simply evaluating differences in observational technique. TLS will work on forest edges but will always be at a disadvantage further into forests versus mobile airborne platforms due to rapid decrease in point cloud density with distance from sensor, analogous to the lambert cosine law, and attenuation of laser penetration though canopy not to mention with slope aspect/slope/curvature/viewshed constraints. Forest edges, including several 10s of m within the forest canopy from the edge have distinctive snow accumulation and ablation energetics (Pomeroy and Gray, 1995; Musselman et al., 2015; Musselman and Pomeroy, 2016). TLS success is always site context/geometry specific while UAV-lidar will not be. Changed to:

 "However, TLS has important limitations to furthering landscape scale understanding of snow processes in forested areas as it is limited by the site specific viewshed and viewing geometry (Deems et al., 2013) and occlusion by forest canopies and low vegetation which decreases point cloud density away from  forest edges (Currier et al., 2019).  It remains an excellent technique for detailed examination of the forest edge snow environment."

**Line 90:** Could add that (Zheng et al., 2016) lidar to understand vegetation processes effect on snow. They particularly note bias that might occur due to tree wells. (Currier & Lundquist, 2018) used lidar to understand the snow-vegetation interactions in multiple climates. (Mazzotti et al., 2019) also used airborne lidar data to improve the understanding of snow depth related to the forest in Colorado and Switzerland.
Have added these references.

**Line 190:** I would mention here that the code is provided on your github page. Great job with providing this.
Have added a reference to github at start of data processing section.

**Line 205:** Trees typically are taller than 50 cm. Most people consider a tree to be at least 2 m tall. Why did you choose 50 cm? This is inconsistent with what the caption shows in Figure 4.
This was a typo and has been corrected.  Classes for vegetation height bins are open <0.5m, shrub 0.5 – 2m, and tree >2m.

**Line 230:** What is estimated and what is observed? I'd say UAV-derived Snow Depth and Snow Depth Probe Manual Observations, or something more specific.
Caption text for Figure 5 has been update to be clearer on what is observed and what is estimated.

**Line 235:** Yes, the reported error metrics are inflated when moving into the forest. It'd be worthwhile mentioning that the sample size is much less.
Have added a sentence to express this.
"The sample size of snow depth probe observations is smaller for vegetation sites than open sites has implications for error metrics –outliers will have greater weight."

Some lidar points do great. In the methods the GNSS mentions a ±2.5 cm accuracy, how was that determined. Is it possible that this is inflated when in the forest? If not, mention that. Are these errors from how the point cloud was processed and points were classified? Is ±2.5 cm true for both horizontal and vertical accuracy?
The accuracy is reported by the Leica GS16 for each point at time of observation.  It is computed from signal quality on the controller as the 3D uncertainty between the base and rover.  Only points in the forest that were able to resolve the RTK solution were collected – there is no decrease in accuracy for the forest/non forest data analyzed. Have added the following to section 2.2.3 in methods: ".  The 3D uncertainty of the relative position between the base and rover was computed in real-time to be < ±2.5cm accounting for errors in signal strength, satellite coverage, and instrument precision. RTK signal quality can degrade in forests but only points with fixed RTK solutions were used in this analysis so all survey points are of equal quality irrespective of vegetation cover."

**Line 238:** I'd start a new paragraph when introducing the error metrics with SfM.
Separated

**Line 245:** The authors should be using Digital Terrain Models instead of Digital Surface Models throughout.

I agree that DSM may not be appropriate here as its definition implies that it is the top of the surface whether that be the soil surface in open areas or the top of the canopy in forested areas.  A DEM is closer to our meaning in that it is a bare-surface raster grid, with trees and vegetation excluded, referenced to a vertical datum.  A DTM on the other hand has various definitions, some of which are incompatible with what we are describing in this paper:

1) DEM can be synonymous with DTM in some countries
   https://gisgeography.com/dem-dsm-dtm-differences/
2) In the US and other countries, a DTM is not a DEM, but is a vector data set composed of regularly spaced points and natural features such as ridges and breaklines. A DTM augments a DEM by including linear features of the bare-earth terrain. https://gisgeography.com/dem-dsm-dtm-differences/
3) DTM: bare-earth representation with irregular spaces between points (non-raster). Behrendt, R. Introduction to LiDAR and forestry, part 1: a powerful new 3D tool for resource managers. The Forestry Source, p. 14-15, set. 2012.

DTM is an acronym with various definitions that may complicate its application here as we are considering both bare ground and snow surfaces beneath a forest canopy.  We feel that it is more appropriate to call these "snow DEM" and "ground DEM" as I am filtering out vegetation points and focusing on the extracted "bare surface" points. Deems et al. 2013 uses DEM to describe snow and bare ground surfaces.

**Figure 6:** Cool analysis. I would consider adding a black dashed line for 2.5 cm. This plot supports the results of Currier et al. 2019, that the airborne lidar is more likely to penetrate the shrubs than the TLS observations. What's the scientific name for the shrubs found at these locations?

Horizontal black line added at 2.5 cm as a reference in Figure 6.  There are many shrub or similar low vegetation species at these locations.  The Prairie sites have a lot of tall prairie grasses and reeds in the wetland areas along with willow and dogwood shrubs and poplar trees. In the cropland around it will be crop residues/standing stubble of wheat or barley.  In the Fortress mountain site, shrubs are primarily willow but there are others too. The site descriptions have been updated to reflect these vegetation details.

**Figures:** I would change the easting northing to the total number of meters within the domain, or start at 0 and show ticks from 0 m. I don't know the projection information, and if I did the numbers aren't that meaningful. If the location is important, please provide the UTM zone. But still it's a bit annoying to do the subtraction each time to get a sense of scale. I would just make it easier for the readers, if possible. Otherwise the figures are great.

Have changed all the figures to have 0,0 UTM origins consistent to each scene.

**Line 317:** This seems like an appropriate time to re-mention UAV lidars ability to capture tree wells.
Have re-mentioned this here.

**Line 321:** Confusing sentence. Deems reported errors in the forest larger than 14 cm? Why is 14 cm mentioned. Figure 5 reports RMSE of 0.15 and 0.16. Also, in the previous sentence. Studies have masked out the forest? Studies have looked at airborne lidar accuracy in the forest.
These results report error metrics for forest situation that are comparable to airborne lidar for open areas.  Some lidar snow depth errors in the forest are comparable to metrics reported here but it's always hard to have apples- to apples comparisons as "forest" is not a uniform landscape class in terms of structure/density and species with respect to how lidar interacts with it.  The advantage of UAV-lidar is that we can get a much broader range in scan angle so we can improve probability of reaching surface points below the tree crown from "oblique" angles. Have changed text to be:

"This RMSE is comparable to previous efforts with UAV or airborne-SfM and airborne-lidar that have been focussed on mapping the snow depth of open snow surfaces. Applications of airborne-lidar to forested areas report similar errors (Currier et al., 2019; Mazzotti et al., 2019) but the higher flight altitude of airborne platforms and their near nadir perspective limit point densities near tree centres that are necessary to capture tree wells."

**Line 355:** Really cool figure and analysis
Thanks!

**Line 375:** Green polygons look cyan when zoomed out, might choose a different color. Furthermore, the near infrared data seemingly comes out of nowhere – maybe provide some more context within the section for it and why it needs to be mentioned. Provide a citation for NIR serving as a proxy for albedo.

I have removed the NIR figure and discussion of its data as it is beyond the scope of the paper and needlessly complicates the story.

**Line 435:** "The accuracy and resolution demands mean that bare surface classification techniques suitable for airborne platforms that efficiently resolve topography and hydrography at watershed scales from last returns will be unsuitable for resolving the snow depth around a particular shrub from a dense point cloud for example" The paper did not show that using the last returns was unsuitable. The classification technique used something similar to last returns. Previous studies have showed using the last returns resulted in a generally unbiased snow depth estimate, and provided a reasonable approximation of the variability. I am not sure what this sentence is attempting to say.

A long winded way to say that where there are dense shrubs the last returns will not necessarily be the snow or ground surface and therefore last-return methods will not be

appropriate.  This is clarified as "Where there are dense shrubs, the last returns will not necessarily be the snow or ground surface and therefore last-return methods common to airborne applications will not be appropriate"

**Line 465:** A discussion referencing the difficulties with modeling in Mark Raleigh's paper seems appropriate and a better citation then Tom Painters 2016 paper. Furthermore, when mentioning snow pack density variability, mentioning Karl Wetlaufer's paper seems appropriate (Raleigh & Small, 2017; Wetlaufer et al., 2016).

These are much more appropriate references- thanks.

**Line 479:** "The UAV-lidar metrics consistently exceed the UAV-SfM metrics and are better than previously reported results in the airborne-lidar and UAV-SfM literature." This isn't true. Metrics are similar but not better than. Please note line 69.
Have rewritten this sentence. To be "The UAV-lidar performance consistently exceeded the UAV-SfM performance and was better than previously reported results in the airborne-lidar and UAV-SfM literature"

References Currier, W. R., & Lundquist, J. D. (2018). Snow Depth Variability at the Forest Edge in Multiple Climates in the Western United States. *Water Resources Research, 54,* 1–18. https://doi.org/10.1029/2018WR022553
Currier, W. R., Pflug, J., Mazzotti, G., Jonas, T., Deems, J. S., Bormann, K. J., et al. (2019). Comparing aerial lidar observations with terrestrial lidar and snow-probe transects from NASA's 2017 SnowEx campaign. *Water Resources Research,* 1–10. https://doi.org/10.1029/2018wr024533
Hedrick, A. R., Marks, D., Havens, S., Robertson, M., Johnson, M., Sandusky, M., et al. (2018). Direct Insertion of NASA Airborne Snow Observatory-Derived Snow Depth Time Series Into the iSnobal Energy Balance Snow Model. *Water Resources Research, 54,* 8045–8063. https://doi.org/10.1029/2018WR023400 Mazzotti, G., Currier, W. R., Deems, J. S., Pflug, J. M., Lundquist, J. D., & Jonas, T. (2019). Revisiting Snow Cover Variability and Canopy Structure Within Forest Stands: Insights From Airborne Lidar Data. *Water Resources Research, 55(7*), 6198–6216. https://doi.org/10.1029/2019wr024898
Raleigh, M. S., & Small, E. E. (2017). Snowpack density modeling is the primary source of uncertainty when mapping basin-wide SWE with lidar. *Geophysical Research Letters, 44(*8), 3700–3709. https://doi.org/10.1002/2016GL071999
Wetlaufer, K., Hendrikx, J., & Marshall, L. (2016). Spatial heterogeneity of snow density and its influence on snow water equivalence estimates in a large mountainous basin. *Hydrology, 3(*1). https://doi.org/10.3390/hydrology3010003
Zheng, Z., Kirchner, P. B., & Bales, R. C. (2016). Topographic and vegetation effects on snow accumulation in the southern Sierra Nevada: A statistical summary from lidar data. *Cryosphere, 10(*1), 257–269. https://doi.org/10.5194/tc-10-257-2016

---

## Author Comment (AC2) · 28 Apr 2020

Reviewer 2
Paper Summary:
The authors compare two relatively new methodologies for using UAVs for mapping snow depths in forested and open prairie environments with in situ ground validation GNSS surveys. They present a very thorough analysis involving an impressive collection of data from 19 unique survey dates from two distinct environments over the course of a single winter season. The time and effort taken to plan, collect, and process such a comprehensive dataset cannot be overstated! The results of the comparison on the ability of both the UAV-lidar and UAV-SfM to estimate snow depths are not necessarily new, but to my knowledge, they have not been compared as extensively with both the successes and failures of both methodologies clearly presented. In open environments, the UAV-lidar and UAV-SfM snow depth mapping capabilities are similar, but in vegetated areas, the UAV-lidar methods excel by having the ability to penetrate through vegetation and measure sub-canopy snow depth. However, in densely vegetated, tight canopy environments, even the UAV-lidar mapping method cannot penetrate the canopy and therefore cannot produce reliable snow depth estimates. An added benefit of using the UAV-lidar over UAV-SfM for snow depth mapping is the insensitivity of the lidar to homogeneous surface conditions and variable/poor solar illumination, both of which contribute to substantial errors in UAV-SfM mapping. In-addition, the increased vertical accuracy of the UAV-lidar sensors can be used to better detect patterns in snow distribution and depth previously not obtainable over basin-wide study sites in complex landscapes. The authors do a nice job at presenting their findings in a well-written manner using suitable figures. As an added bonus, the authors also discuss the cost difference between the UAV measuring methodologies, and calculate a metric that assigns a dollar value to each centimeter of improved RMSE between methods. This cost analysis is of interest, but probably has less relevance for the future, as the price for the type of equipment used in this study continues to decrease dramatically year-by-year. I recommend the publication of this paper pending minor revisions addressing the suggested comments and technical edits.
A PDF supplement has also been uploaded that contains all the suggested edits/comments. In the technical edits section, this PDF supplement has all changes highlighted in **BOLD.**
An example of the suggested changes to Figure 7a has also been uploaded as Figure 1 – Slide 1.JPG. This example figure provides a visualization of the changes being suggested for Figures 7a, 8a, 9a (applies to General Comment at Line 270/295/300).

First, thank you for this detailed review and you will find our responses in red below the corresponding comment.

General Comments:
Line 59 – 'differencing snow-covered (hereafter snow) and snow-free (hereafter ground)…' Double check terminology throughout paper for consistency. The following different term are used: bare-ground, bare ground, ground, surface, bare surface. Personally – I like the use of the term bare-ground.

We refer to 'surfaces' which can be either snow or snow-free. "Bare' refers to points left after vegetation point removal for either a snow or snow-free surface. 'Ground' implies a snow-free surface. We have edited the paper to make the terminology more consistent, following these rules throughout.

Line 59 – 'Digital Surface Models (DSMs)' I think you are actually referring to the Digital Terrain Models (DTMs). Change this reference throughout the paper.

I agree that DSM may not be appropriate here as its definition implies that it is the top of the surface whether that be the soil surface in open areas or the top of the canopy in forested areas. A DEM is closer to our meaning in that it is a bare-surface raster grid, with trees and vegetation excluded, referenced to a vertical datum. A DTM on the other hand has various definitions, some of which are incompatible with what we are describing in this paper:
1) DEM can be synonymous with DTM in some countries https://gisgeography.com/dem-dsm-dtm-differences/
2) In the US and other countries, a DTM is not a DEM, but is a vector data set composed of regularly spaced points and natural features such as ridges and breaklines. A DTM augments a DEM by including linear features of the bare-earth terrain. https://gisgeography.com/dem-dsm-dtm-differences/
3) DTM: bare-earth representation with irregular spaces between points (non-raster). Behrendt, R. Introduction to LiDAR and forestry, part 1: a powerful new 3D tool for resource managers. The Forestry Source, p. 14-15, set. 2012.

DTM is an acronym with various definitions that may complicate its application here as we are considering both bare ground and snow surfaces beneath a forest canopy. We feel that it is more appropriate to call these "snow DEM" and "ground DEM" as I am filtering out vegetation points and focusing on the extracted "bare surface" points. Deems et al. 2013 uses DEM to describe snow and bare ground surfaces.

Line 134 – 'flight parameters to maximise mapping efficiency were set to…' What about limiting the scan angle? The Riegl lidar can scan 360 degrees, what level of off nadir scan angle did you limit the data collection/processing to and why?
The Riegl scanner does scan in a 360° configuration. While data can be limited to specific scan angles at collection it was not limited in our application, as there is no increase in performance/accuracy to do so – the mirror is rotating the full 360°. The scan angle was not limit in processing the data either. The laser is relatively low powered and we have found that returns at angles shallower than 70° from nadir are rare. Hence, we did not limit the available data to perform our analysis – any points available were used to optimize the surface feature extraction.

Line 135 – '100 m flight altitude above the surface…' Did the mission planning software make use of terrain following mode to ensure consistent flight altitude above ground? If so, what source of terrain information did you use?

Yes used terrain following with respect to a SRTM DEM. Have added "The UgCS flight control software was used to generate flight paths with these parameters and terrain following with respect to an underlying SRTM DEM"

Line 148 – I deleted the term differential: differential GNSS corrections (code-based) are significantly less, accurate than RTK/PPK/PPP (carrier phase methods) – I suspect even though the Leica GS16 unit is DGPS capable, you used the more accurate carrier phase correction methods.

Correct, we were using the carrier phase methods. 'differential' has been removed

Line 150 - suggest removing the term 'random within the survey areas and' if the transects were also selected to most efficiently survey the greatest variety of vegetation types.
Agreed – we have removed that text.

Line 152 – 'provided a real-time-kinematic (RTK) survey solution …' While conducting your manual surveys did you make use of the RTK capabilities – or did you post-process the rover data as indicated at line 153?

The difference between the rover and base was established during the surveys with RTK. Because the base position was not known in advance to the surveys the RTK observed rover positions needed to be adjusted to absolute locations in post processing once the base position was established through the PPP step – an offset needed to be calculated and applied to survey points. In post processing the only adjustment was made to the base position not the relative rover-base positioning. This is clarified as:
" Post-processing with Leica Infinity software (version 2.4.1.2955) established the absolute positions of the rover points by maintaining the RTK rover-base position but adjusting the base station absolute location to that established by the PPP tool."

Line 152 – 'accuracy of < ±2.5cm.' Can you provide a reference for this?
Not shown here but the uncertainty is computed in real-time as part of the RTK and PPP based solutions see below and was consistently less than of < ±2.5cm –not based on a reference.

Line 154 – '(https://webapp.geod.nrcan.gc.ca/geod/tools-outils/ppp.php)' Add this website to the references section
Have added this to the references.

Line 154 – 'absolute base station location.' How long did you collect your raw GNSS data for and what were the PPP computed standard deviations for the base station locations? Did you always use the same base station location for every flight?

Due to the logistics of conducting campaigns at multiple sites, raw GNSS data was only logged when we were on site with different tripod setups.  Therefore logging varied in duration between 2.5 and 9 hours.  The PPP computed standard deviations were

consistently less than 2cm –often better. For simplicity the uncertainty of the survey solution was presented to be ± 2.5 cm.  This value is based propagating a conservative uncertainty of the PPP based solution 2cm and the RTK solution off 1.5cm.  sqrt (2^2+1.5^2)=2.5. We have updated section 2.2.3 to reflect this.

Line 174 – '<2.5 cm.' Do you mean +/- 2.5 cm as mentioned earlier in the text? Is this value based on the specs of the Leica GS16 GNSS survey equipment or was it based on the PPP online standard deviations? How did you obtain this value?
See comment above

Line 181 – '< ±2.5 cm' Same comment as above? Is this value based on the specs of the Leica GS16 GNSS survey equipment or was it based on the PPP online standard deviations? How did you obtain this value?
See comment above

Line 205 – 'vegetation height (open <0.1 m, shrub <0.5 m, and trees >0.5 m)…' These values differ from what is in the Figure 4 caption. Which vegetation height classes did you use, and how did you choose the class heights?
The caption for figure 4 and text were slightly incorrect due to relics of an earlier edit. Vegetation height classes were open <0.5 m, 0.5 m ≥ shrub ≤2 m, and trees >2 m. Vegetation classes were selected with a simple metric to differentiate vegetation based on the height data at hand.  There will be variability in shrub heights, but for simplicity we used 0.5m and 2m thresholds as they were consistent with field observations at the various sites and thresholds previously reported in the snow hydrology literature which ranged from 0.5m to 3m in Marsh et al. (1997) and 0.3m to 2m in Rasouli et al. (2019).

Marsh, P., Pomeroy, J.W., Pietroniro, A., Neumann, N., Nelson, T., 1997. Mapping Regional Snow Distribution in Northern Basins Inuvik Area. Saskatoon, Saskatchewan. http://citeseerx.ist.psu.edu/viewdoc/download?doi=10.1.1.712.6847&rep=rep1&type=pdf

Rasouli K., Pomeroy J.W., and Whitfield P.H. (2019) Are the effects of vegetation and soil changes as important as climate change impacts on hydrological processes? Hydrology and Earth System Sciences: 23, pp. 4933-4954 DOI: 10.5194/hess-23-4933-2019

Line 223 – I deleted reference to RTK - In line 55 you indicate the rover survey points were post-processed, therefore I am assuming you used a PPK GNSS solution here?

RTK with a post processing of the base position to account for PPP.  See comments above.

Line 230 – 'points extracted from the point clouds or interpolated surfaces…' This sentence is confusing. It is unclear whether you extracted the UAV snow depth values from the point clouds or the interpolated DSMs? Which one was it?

Caption was in error and is corrected as "Plots are segmented for vegetation class (rows), sites (columns) and observation method (colours)."

Line 256 – Figure 6 - Please add to the caption a description of which metrics are visualized by the whiskers of the boxplots.

Have added: "Median is indicated by the line inside the box, the upper bound is the 75th percentile and the lower bound is the 25th percentile and whiskers represent the range of values beyond the box."

Line 266 – 'The noisy UAV-SFM points in the middle of the slope challenge the snow surface extraction even without the presence of vegetation leading to an underestimation of the snow surface.' Do you have any idea on why the SfM product detected something in the open areas on the slope? Why does it lead to an underestimation of snow in this area? Based on the Figure 7a cross-section it looks like the UAV-SfM red points are equal to or above the green lidar points. Why did the interpolation go so low? Did the interpolation treat missing points as 0 or bare ground values?

Looking closer at the UAV-SfM noise in Figure 7a there was some vegetation mid slope near to the transect. The lidar was able to differentiate it well but the SfM-generated vegetation points occupied a larger space and intruded on the transect line. Therefore when vegetation was removed it led to a gap in the UAV-SfM point cloud at this point in the transect, which when interpolated through led to the underestimation in the snow surface. When gaps are present the interpolations are sensitive to edge points which tend to have poor quality and therefore challenge the validity of the resulting surface.

Have added: "These vegetation points occupied a larger space than the UAV-lidar and intruded on the transect line. Therefore, vegetation removal from this point in the transect led to a gap in the UAV-SfM point cloud, but not the UAV lidar point cloud. Interpolating through the gap in the UAV-SfM point cloud resulted in underestimation of the snow surface."

Line 270/295/300 – Figure 7-8-9 - Suggest using shaded/transparent colour bars on plot a) to indicate the extent of the tree features. This will help highlight the tree well extent and how the UAV-SfM interpolation result in deeper snow values across these features (I have uploaded an example Figure of 7a. that illustrates what I am trying to describe – Slide 1.JPG). Suggest using a more obvious colour in Figure b) for highlighting the SfM only classes. Suggest trying to match the tone of colours in Figure c) to more closely match that used in Figure b). Making the open areas a little bluer, and again highlighting the SfM only points in a more obvious colour. Figure 7b It sort of looks like the SfM only class occur near the edges of the study area in a just a couple areas. Is this related to steeper scan angles at the edge of the study site, perhaps coupled with steep terrain?

Have modified the figures to have polygons outlining areas of interest in the cross sections and changed the colour scheme to more clearly show differences in point

coverage.  The SfM-only points occurred on the edges of the domain as this was nearing the edge of the lidar flight area (less overlapping scan areas reduces the point density and therefore reduces number of ground points)

Figure 9c) I suggest mentioning in the figure caption that the large dark areas of no lidar points represent the extent of the melt water ponds.
The figure caption is rather large already so prefer to leave this discussion in the text.

Line 288 – the negative UAV-SfM snow depth estimates discussed here are explained at lines 443-450. Perhaps also providing further explanation here might be helpful.
To simplify the results section would suggest that this explanation fits better into the discussion section as it is.

Line 316 – In the example of 7a, the interpolation resulted in erroneously deep snow depth estimates. This will not always be the case and in some instances can result in underestimations depending on the season, elevation, forest type, etc. Many studies have highlighted the differences in snow depths/characteristics between open/forested sites that will influence these interpolation errors. I think providing some further explanation on the type/magnitude of interpolation errors that may occur when using UAV-SfM techniques would help strengthen your findings/statement here.

Have added: "Open areas will have greater snow depths than forest areas (Troendle 1983; Swanson et al., 1986;  Pomeroy et al., 2001; Mazzotti et al., 2019;) meaning UAV-SfM solutions, or any approach which requires interpolation of point cloud gaps beneath trees, will overestimate snow (Zheng et al., 2016)."

Line 318 – 'major improvement on previous attempts.' Can you provide some context on what is considered a major improvement, including references to previous studies/RMSEs?

Have removed "major" as that is an unquantifiable adjective.  Have modified it to be "The ability of UAV-lidar to map snow-depths, with and without canopy cover, and capture tree wells with RMSE's ≤0.15 m is an improvement on previous attempts.  This RMSE is comparable to previous efforts with UAV-SfM (Bühler  et al., 2016; De Michele et al., 2016; Harder et al., 2016), airborne-SfM (Bühler  et al., 2015; Nolan et al., 2015, Meyer and Skiles 2019) and airborne-lidar (Deems et al., 2013; Painter et al., 2016) that have been primarily focussed on mapping the snow depth of open snow surfaces. Applications of airborne-lidar to forested areas report similar errors (Zheng et al., 2016; Currier et al., 2019; Mazzotti et al., 2019) but the higher flight altitude of airborne platforms and their near nadir perspective limit point densities near tree centres that are necessary to capture tree wells."

Line 318 – 'previous efforts…' Can you provide some references?
Same as comment above.

Line 321 – '0.14 m RMSE (Deems et al., 2013).' Can you provide the actual magnitude of errors previously reported for comparison in the Deem et al., 2013? What is the significance of this 0.14 m RMSE?

Have removed this 0.14 m RMSE per comment from Reviewer 1

Line 342 – 'intermittent precipitation totaling approximately 100 mm' How was this determined/measured? What kind of uncertainties are associated with this reported precipitation value. I also want to confirm that you mean 10 cm of snow? This seems low for mountain snow.

There are a number of precipitation gauges (Geonor and Pluvio) within the Fortress mountain research basin. I say 'approximately' as this was an approximation of the raw storage gauges signals as the data QA/QC and undercatch corrections were beyond the scope of this project. And yes I do mean that this is approximately 10 cm of snow. It is low for a mountain situation but 2019 was a low snow year in this area and the February to April interval this is reflecting was a cold, dry period without any major snowfall events.
Have added; "measured at storage gauges at the study site"

Line 350 – 'and development of a tree well in the middle of the transect. The Figure 10b transect demonstrates the lack of wind redistribution in the canopies relative to the Figure 10c transect on the ridgeline.' It is unclear where the development of the tree well is highlighted/visible in Figure 10b. It also unclear how Figure 10b demonstrates the lack of wind re-distribution in the canopies. Please provide more detail here.

Have highlighted the tree wells with orange polygons in figure 10b. Have added the following to clarify the comment on demonstrating a lack of wind redistribution in the forest area. "The Figure 10b transect demonstrates the lack of wind redistribution in the forest; snow accumulation was consistently observed to be ≤ precipitation over the transect, versus the Figure 10c transect on the ridgeline, where the accumulation in the lee slope greatly exceeded the observed precipitation."

Line 366 – 'In contrast UAV-SfM struggled with sensing snow depths in the short shrubs on the edges of wetlands.' This sentence contradicts the results displayed in Figure 5, which illustrated that the UAV-SfM had lower RMSE in the shrub class compared to the UAV-lidar. It also does not support the discussion starting at Line 286 and expanded at Lines 443-450, which discusses the challenges that BOTH lidar and SfM face in trying to measure below the canopy in dense shrub vegetation.

This sentence needed to be a bit more nuanced. This is not a comment on the RMSE differences and should not have highlighted the shrubs in particular rather this was based on the fact that there is a higher point cloud density for the lidar versus SfM in wetland areas. This is clarified as "In contrast UAV-SfM struggles with sensing snow depth on the edges of wetlands as seen by the concentration of lidar only areas at the wetland in the Rosthern study area (wetland area highlighted by red polygon in Figure 8b).

Line 467 – 'Observational approaches are also a challenge as typical in situ measurements are destructive, limited in extent, and often too limited to develop robust relationships of depth versus density at the small scales needed (Kinar and Pomeroy, 2015a; Pomeroy and Gray, 1995).' The methods developed by Proksch et al., 2015 do provide a method for measuring snow density at a much smaller scale applicable for these process-scale studies. The Proksch et al., 2015 methods have been recently rigorously applied to a set of snow on sea ice measurements by King et al., 2020, highlighting the ability to document the local-scale variations in snow density relatively quickly over larger spatial extents.

Proksch, M., Löwe, H. and Schneebeli, M., 2015. Density, specific surface area, and correlation length of snow measured by high-resolution penetrometry. Journal of Geophysical Research: Earth Surface, 120(2), pp.346-362.

King, J., Howell, S., Brady, M., Toose, P., Derksen, C., Haas, C., and Beckers, J.: Local-scale variability of snow density on Arctic sea ice, The Cryosphere Discuss., https://doi.org/10.5194/tc-2019-305, in review, 2020.

These methods while very interesting and small scale are still destructive sample methods which means that their application to a UAV-based solution to SWE estimation, that captures local and landscape scale density spatial and temporal variability, will be limited. The small sample size and empirical calibration of the micro-penetrometer method results in uncertainty in its application.

This is communicated through a slight edit as " Observational approaches are also a challenge as typical in situ measurements are destructive, limited in extent, and often too limited to develop robust relationships of depth versus density at both the small local and large landscape scales needed (Kinar and Pomeroy, 2015a; Pomeroy and Gray, 1995). Opportunities may be available to pair UAV-lidar with other UAV-borne sensors such as passive gamma ray or snow acoustics (Kinar and Pomeroy, 2015b) to non-destructively develop high spatial and temporal resolution estimates of snow density and ultimately water equivalent.."

Line 474 – 'necessary spatial scales' – Please be more specific on what scales you are referring to.
Have removed this sentence as the scales are mentioned later in the conclusion section.

Technical Comments:
Line 13 – suggest changing to 'measure returns from a wide range of scan angles, **increasing the** likelihood of successfully…'
Changed

Line 51 – suggest changing to 'are valuable automated data sources, but **are spatially limited in extent and can often** suffer from location/elevation bias…'
Changed

Line 53 – suggest changing to 'and so **may** not **be** suitable for snow hydrology calculations or model validations **in forested regions** even though they are often…'
Changed

Line 60 – spelling correction: quality
Changed

Line 62 – suggest changing to 'pulse can be observed with **returns possible from within the canopy and from the sub-canopy ground surface.** In contrast UAV-SfM…'
Changed

Line 64 – spelling correction: variability
Changed

Line 80 – spelling correction: focused
This is correct Canadian English spelling.

Line 87 – punctuation: 'In dense forests, vegetation…'
Changed

Line 90 – suggest changing to 'increase in snow accumulation **over** aerodynamically rough surfaces or **in** sheltered areas **where the wind speeds decrease and snow is deposited** – this includes forest edges…'
Changed

Line 98 – suggest changing to 'varies across complex vegetated landscapes…'
Changed

Line 105 – suggest changing to 'ability of the UAV-lidar and UAV-SfM **techniques for measuring** snow depth in open
Changed

Line 106 - (50.833 N, 115.220 W)
Changed

Line 108 – spelling correction: focused
This is correct Canadian English spelling.

Line 109 – suggest changing to '(Figure 1a – background center)…'
It is already directly identified as the a) panel so will leave as is.

Line 111 – suggest changing to 'alpine ski resort in the 1960's, **but is** currently a limited-use…'
Changed

Line 114 – suggest changing to 'Canadian Prairies were **examined** in this study.'
Changed

Line 117 – correction: remove negative sign if using 'W' to indicate west (51.941 N, 106.379 W) & (52.694 N, 106.461 W)
Changed

Line 125 – Figure 1 caption: suggest changing to 'Figure 1: a) Fortress Mountain Snow Observatory in Kananaskis, Alberta Canada, b) **Rosthern** and c) **Clavet prairie** study locations in Saskatchewan Canada. Data collection was on Fortress Ridge (**background center)** an area of high topographic variability and **a mix of** dense forests and clearings. The Clavet **photo** highlights the **transition zone between the open upland terrain and the lower elevation vegetated** wetland. The Rosthern scene highlights the low vertical relief **of upland areas** and isolated
woodlands amongst cultivated fields.
Changed

Line 155 – suggest changing to 'GS16 rover points to **correct** for the PPP **updated** base station locations **were completed using** the Leica Infinity software…'
This section has been reworked and this edit no longer applies

Line 158 – 'suggest changing to 'To assess the accuracy of the **UAV snow depth measuring** methods, as well as provide insight into the **seasonally evolving** snow **depth/**distribution, **a total of** 19 **flight/manual** surveys were conducted **between all three study sites between** September 2018 to April 2019. These are summarised by date, **surveyed** surface, **UAV** data collected, and corresponding number of **manually surveyed** surface elevation points in Table 1.
Changed

Line 165 – suggest changing to 'difference between a bare ground DSM and a snow **surface** DSM.'
Changed but using DEM rather than DSM

Line 176 – suggest changing to 'Finally, overlapping scan data **from adjacent flight lines are** used to optimise the IMU trajectory, to align the scan lines and reduce the noise of the final point cloud within the RiPrecision tool. **This final step in noise reduction can improve the final product because the 1.5 cm laser data precision is greater than the post processed IMU trajectory accuracy.** (I used the 15mm stated precision of the Reigl sensor presented earlier in the text to get the 1.5cm value here)
Changed and absolutely correct on that last sentence to clarify matters.

Line 193 – suggest changing to 'For **the bare-**ground lidar scans, the height of vegetation…'

Changed

Line 207 – spelling correction: include
Changed

Line 214 – suggest changing to '2.3.6 **Point Cloud Density'**
Changed to 'Point Cloud Coverage' as density has a different meaning.  Here I'm trying to quantify how gappy the bare point clouds are.

Line 221 – suggest changing to '3.1 **Accuracy of UAV-lidar versus UAV-SfM snow depth estimates**
Changed

Line 231 – suggest changing to 'Plots are segmented for points extracted from the point clouds or interpolated surfaces **within each vegetation class** (rows), sites (columns) and observation method (colours).' – See general comments above about clearing up the confusion concerning which product the points were extract from.
Changed

Line 232 – suggest changing to 'The influence of vegetation on **estimating** snow depths **from UAVs can be** directly assessed by…'
Changed

Line 234 – suggest changing to 'Open Prairie and **open** Fortress **RMSE values** are similar (0.09 m and 0.1 m RMSE respectively)…'
Changed

Line 235 – suggest changing to 'equally successful **at** penetrating the open leaf-off deciduous tree canopy at the prairie sites as the closed needleleaf canopy at the Fortress site **based on the similar RMSE values within each site's tree vegetation class.'**
Changed

Line 238 – suggest changing to 'The Open vegetation has a large RMSE range **between sites** (0.1 m in Prairie and 0.3 m in Fortress respectively) while vegetation **class** RMSEs range from…'
Changed

Line 240 – suggest changing to 'UAV-lidar in the prairie Shrub case, **the difference between these techniques is only 0.04 m, which is** within the **+/- 2.5 cm** observational **uncertainty** of **the GNSS** survey equipment **used in this project.**
Changed

Line 247 - suggest changing to 'manual **GNSS** surveys **using boxplots** (Figure 6). **The boxplots in Figure 6 illustrate that** the UAV-SfM snow surface elevations…'
Changed

Line 257 – suggest changing to '3.2 **Point cloud density**'
Changed to "Point cloud coverage" per previous comment.

Line 263 – suggest changing to 'could not reliably return surface points **with a density > 1 pt 0.25 m-2**whilst…'
Changed

Line 263 – punctuation: 'At Fortress, UAV-lidar…'
Changed

Line 265 – suggest changing to 'lack of UAV-SfM sub-canopy points identified within the treed vegetation class results in an interpolated snow surface that is erroneously deep under trees, completely missing the detection of the reduced snow depths which are clearly detected (green line) around the base of the trees by the UAV-lidar.'
Changed

Line 274 – suggest changing to 'c) with **the same** overlain transparent point type classification **colour scheme as shown in b).**'
Changed

Line 276 – suggest changing to 'The predominantly open nature of the Prairie sites demonstrates a minimal difference in **point density** between **UAV-lidar and UAV SfM measurement** techniques. The average **extent of the study domain covered with a point density of > 1 pt 0.25 m2 for** 5 coincident flights at the Prairie sites **was computed, resulting in the** mean coverage of 92% versus 83% **of the study area** for **the UAV-lidar and** UAV-SfM **respectively.**
Changed

Line 281 – suggest changing to 'These gaps in the **UAV-SfM** point clouds are interpolated and therefore will represent…'
Changed

Line 287 – suggest changing to 'both lidar pulses and SfM solutions interpret the vegetation surface as the **top of the bare-**ground **or snow** surface and therefore **little difference exists between these two DSMs during all measurement periods.** An additional challenge of **using the** UAV-SfM **techniques** is that large gaps **in points** appear beneath the tall wetland edge vegetation due to the inability to penetrate the sub-canopy, as visualized **in the cross-sections** of Figure 8a and 9a**,** where the estimated UAV-SfM snow surface is below the UAV-lidar ground surface.'
Changed

Line 316 – suggest changing to 'Sub-canopy snow depth mapping with UAV-SfM therefore becomes an exercise in **interpolating snow depth values observed in open** areas **without** vegetation **to areas with dense vegetation,** rather than sensing the actual snow depth under the canopy.'

Changed

Line 322 – suggest changing to '4.2 **Bare-ground point cloud density is critical**'
Ground in this case can be either 'ground' or snow so 'surface remains more appropriate.

Line 323 – suggest changing to 'The **increased** point **density** of UAV-lidar…'
Not so much density as lack of gaps aka coverage.  Changed to "The increased continuous point coverage of UAV-lidar"

Line 325 – suggest changing to 'The point cloud cross-sections **illustrated** in Figure 7 emphasize **these findings, highlighting the** wider gaps in the UAV-SfM point cloud beneath individual trees that require interpolation **over longer distances resulting in greater potential for error.**' (The lidar data also requires interpolation)
Changed

Line 332 – suggest changing to 'In contrast, **mountainous regions** have much more complex topography…'
Changed

Line 337 – suggest changing to 'continuous bare-**ground** point cloud coverage.'
Ground in this case can be either 'ground' or snow so 'surface remains more appropriate.

Line 338 – suggest difference word choice for: foreshadow
Changed to 'Two examples are presented here to exemplify analyses the possible with UAV-lidar'

Line 340 – suggest changing to 'Differences between open and forest snow cover processes can be **explored** by **examining** the difference in snow depth…'
Changed
Line 342 – suggesting changing to 'UAV-**lidar measured** change in snow depth visualizes…'
Changed

Line 343 – suggest deleting line: 'The upper, open terrain clearly demonstrates the influence of blowing snow redistribution' because this sentence is ambiguous.
Line deleted

Line 343 – suggest changing to 'In the Figure 10c transect **cross-section** there was accumulation of up to 2 m over the **September-April time** period on lee slopes, whilst the upper windswept portions of the ridge demonstrate snow erosion **between February and April.**"
Changed

Line 346 – suggest changing to 'The dynamics and extents of blowing snow sources **(grey/red)** and sinks **(blue)** are clearly visualized in **10a, which closely match the findings of** Schirmer and Pomeroy (2019) using SfM **for this same study region.**
Changed

Line 347 – suggest deleting line: 'Considering the forest slope brings out features that UAV-SfM cannot observe.' Because this sentence appears as a fragment
Deleted sentence

Line 349 – suggest changing to 'there is a general decline in snow depth **from February to April** (due to melt on the south facing slope).'
Changed

Line 360 – suggest changing to 'wind-blown snow from **open** upwind sources and are typically associated with…'
Changed

Line 366 – suggest changing to 'Areas that the UAV-lidar was able **to** measure correspond to areas…'
Changed

Line 390 – suggest changing to 'This gradient in dust and albedo **is likely associated with** the increases in snowmelt rates **observed** downwind of the grid road.'
Section has been reworked.

Line 405 – suggest changing to 'UAV-lidar, relative to UAV-SfM, provides **the ability to measure** snow depth below vegetation…'
Changed

Line 408 – suggest changing to 'and cheaper **equipment,** subscriptions to virtual reference station networks if available in the study area **(requires only a rover and not a base station),** or equipment rentals are all viable alternatives to lower costs.'
Changed

Line 410 – suggest changing to 'The main cost difference **between UAV-lidar and UAV-SfM platforms** is therefore in terms of the **UAV** sensor **payload.'**
Changed

Line 412 – suggest changing to 'like consumer grade UAVs (DJI Phantom 3 < $2,000 CAD), **to** more expensive options like…'
Changed

Line 413 – suggest changing to 'Current integrated lidar systems suited to **UAV** snow mapping'
Changed

Line 423 – suggest changing to 'In contrast, **most current** UAV-lidar **configurations** need larger platforms that require more cycles of large battery sets to cover similar areas, which represents a **logistical** challenge in **keeping the batteries warm and charged in** cold and remote areas.'
Changed

Line 428 – suggest changing to 'Despite the lower **initial purchase** cost and **longer flight endurance,** the errors and artefacts that UAV-SfM **measuring techniques** introduce in sub-canopy **snow depth measurements,** as detailed in sections 4.3.1 and 4.3.2, **suggest that UAV-SfM is not able to directly measure snow depth** in **densely vegetated environments.'**
Changed

Line 434 – suggest changing to 'Precise classification of surface points from snow and ground scans **are** needed to resolve…'
Changed

Line 435 – suggest changing to 'The accuracy and resolution demands **are such** that bare-**ground** surface classification techniques **developed** for airborne platforms to resolve topography and hydrography at watershed scales from **lidar** last returns **may** be unsuitable for resolving snow depths.'
Have changed this sentence with respect to Reviewer 1 comments

Line 438 – suggest changing to 'filtering tools and associated parameters to **be able to reliably detect the sub-canopy bare-ground surface and** achieve desired quality…'
Changed

Line 441 – spelling correction: 'large-scale'
Changed

Line 448 – suggest changing to 'the areas of negative snow are limited to areas where snow depth is **relatively** shallow **in comparison to the** deep snow in the wetland edges.'
Changed

Line 452 – suggest changing to 'snow depth estimation in **these** hydrologically significant snow accumulation areas.'
Changed

Line 453 – suggest changing to 'ground surface, but **current** sensors **with these characteristics** may exceed **the payload capacities** of most UAV platforms. Advances in bare surface classification/**filtering** software…'
Changed

---

## Author Response (AR2)

**Author Response**

This document provides a point-by-point response to the editor corrections followed by a marked-upversion of the revised manuscript. Responses are in red text.

5 Phillip Harder May 6, 2020

Line 141: remove 'following'

This sentence has been modified to be "The UgCS flight control software (SPH Engineering, 2020) was used to generate terrain following flight paths with respect to these parameters and an underlying SRTM DEM".

Line 252: change to '...is smaller for vegetation sites than Open sites, which has implications..." Corrected

15 Line 290: 'Should 'orange polygon' be in brackets?

**Orange polygons is now in brackets**

Line 389: 'snow accumulation was consistently observed to be ≤ precipitation over the transect' Presumably this is because of interception? Perhaps specify this? Replace the < symbol with

**20 **text**.**

10

Have changed/added the following to the sentence to be clearer : "snow accumulation was consistently observed to be less than precipitation over the transect due to interception losses"

In addition have changed Figure 4 so that UTM coordinates are now with respect a study area origin not UTM zone to be consistent with all other figures.

**Improving sub-canopy snow depth mapping with unmanned aerial vehicles: lidar versus structure from motion techniques**

Phillip Harder1, John W. Pomeroy1, and Warren D. Helgason1,2

1Centre for Hydrology, University of Saskatchewan, Saskatcon, Saskatchewan, Canada

[revised manuscript text omitted]

- 170 This ground speed, 100 in Fight and de above the surface, with parallel fight miles 50 in apart. The eges fight control software (SPH Engineering, 2020) was used to generate flight paths with these parameters and terrain following with respect to an underlying SRTM DEM. The UgCS flight control software (SPH Engineering, 2020) was used to generate terrain following flight paths with respect to these parameters and an underlying SRTM DEM. Flight times are conservatively limited to 15 minutes. The generated UAV-lidar point clouds have densities of approximately 75 points per square metre (pt m-2).

**175 2.2.2 Structure from Motion systems**

Coincident surface mapping with SfM used imagery collected by EbeeX or Ebee+ fixed wing UAV platforms with SODA RGB cameras from Sensefly (Figure 2b). The longer flight times, up to 70 minutes, associated with a lightweight payload on a fixed wing platform allowed for efficient mapping of large areas. Overlap parameters were generally 80% for the longitudinal and 65% in the lateral axes. Flight altitudes of 120 m above the surface provided a ground sample distance of 2.8 cm with the SODA camera, which was used on both EbeeX and Ebee+ platforms. The generated UAV-SfM point clouds have densities of ~ 110 pt m-2.

180